



# Mid-Holocene monsoons in South and Southeast Asia: dynamically downscaled simulations and the influence of the Green Sahara

Yiling Huo[1], William Richard Peltier[1], and Deepak Chandan[1]

[1] Department of Physics, University of Toronto, Toronto, M5S 1A7, Canada

*Correspondence to*: Yiling Huo (yhuo@physics.utoronto.ca)

**Abstract.** Proxy records suggest that the Northern Hemisphere during the mid-Holocene (MH), to be assumed herein to correspond to 6,000 years ago, was generally warmer than today during summer and colder in the winter due to the enhanced seasonal contrast in the amount of solar radiation reaching the top of the atmosphere. The complex orography of both India and Southeast Asia (SEA), which includes the Himalayas and the Tibetan Plateau (TP) in the north and the Western Ghats mountains along the west coast of India in the south, renders the regional climate complex and the simulation of the intensity and spatial variability of the MH summer monsoon technically challenging. In order to more accurately capture important regional features of the monsoon system in these regions, we have completed a series of regional climate simulations using a coupled modeling system consisting of the University of Toronto version of the Coupled Climate System Model version 4 (UofT-CCSM4), the Weather Research and Forecasting (WRF) regional climate model and the 3D Coastal and Regional Ocean Community model (CROCO) to dynamically downscale MH global simulations constructed using UofT-CCSM4. In the global model, we have taken care to incorporate Green Sahara (GS) boundary conditions in order to compare with standard MH simulations and to capture interactions between the GS and the monsoon circulations in India and SEA. In both the global and the regional models, the response of the South Asia (SA) and SEA monsoons to MH orbital forcing is intensified and accompanies lower surface temperature which is likely related to the increased reflectance of shortwave flux at high levels from the greater cloud cover. Comparison of simulated and reconstructed climates suggest that the dynamically downscaled simulations produce significantly more realistic anomalies in the Asian monsoon than the global climate model, although they both continue to underestimate the inferred changes in precipitation based upon reconstructions using climate proxy information. Monsoon precipitation over SA and SEA is also greatly influenced by the inclusion of a GS, with a large increase in particular being predicted over northern SA and SEA, and a lengthening of the monsoon season. Data-model comparison with downscaled simulations outperform those with the coarser global model, highlighting the crucial role of downscaling in paleo data-model comparison.



## 1 Introduction

The climate over both South and Southeast Asia (SA and SEA) is marked by distinctly different surface temperature, lower tropospheric wind direction and precipitation between summer and winter, resulting in a wet summer and a dry winter. During summer, warm and moist air originating from the Indian Ocean is transported northeast towards the Indian subcontinent and the Indochina peninsula where the air-mass organizes into the South and Southeast Asian monsoons (SAM and SEAM, respectively) circulations. The strengths of these circulations and their onset, maintenance and withdrawal are to a significant extent influenced by the contiguous Tibetan Plateau (TP) that serves an elevated heat source for the atmosphere in summer that intensifies the thermal contrast between the continent and ocean in the region influenced by the Asian monsoons (Wu et al., 2007). The intensity and variability of precipitation from these two monsoonal systems greatly affect the population inhabiting this region, which relies upon the monsoon rainfall as their primary freshwater source (Turner and Annamalai, 2012). Understanding changes in monsoon circulation and precipitation on different time scales is thus of great scientific and socioeconomic importance.

As with any monsoon circulation, seasonal and latitudinal differences in the incoming solar radiation and the land-ocean thermal contrasts caused by the heat capacities difference between the land and sea, which cause a phase lag between the period of peak surface heating between land and sea, are fundamental to the formation of SAM and SEAM (Webster et al., 1998). Whereas this external forcing is essential for initiating SAM and SEAM, internal feedback processes, which have been recognized as changing over the recent decades (Mishra et al., 2018; Endo et al., 2009), play a crucial role in amplifying their strength. Since monsoon systems are characterized by a strong temporal variability (Ding, 2007; Kutzbach, 1987; Prell and Kutzbach, 1987), the limited modern observational datasets are insufficient to constrain the expected future response of these systems to climate change. Additionally, projections of monsoon evolution from climate modeling of future conditions remain highly uncertain (Huang et al., 2020). Therefore, analysis of past changes of monsoon circulation systems under significantly different forcing conditions for both SAM and SEAM is potentially of great value in contributing to the understanding of these monsoons and their expected future changes.

The mid-Holocene (MH), approximately 6000 years ago, is characteristic of a typical interglacial period during which the insolation distribution was significantly different from that of the present-day (Fig. 1). For this reason, the period offers an excellent opportunity for investigating the response of the Asian monsoon to the



precessionally enhanced Northern Hemisphere insolation which in the latitudinal range of these monsoons induces a top-of-the-atmosphere insolation increase of ~20 W m$^{-2}$ in summer due to differences in the Earth's orbital

configuration compared to present-day (Laskar et al., 2004; Berger, 1978). Nearly all of the earliest reconstructions for the MH Asian monsoons were based on palaeoceanographic evidence, including distribution of planktonic fauna (Hutson and Prell, 1980; Prell, 1984a, b; Cullen and Prell, 1984) and wind-transported pollen (Prell and Van Campo, 1986) in deep sea sedimentary cores obtained from the Indian Ocean. Subsequently, much progress in studying these paleomonsoons has been achieved through continuing studies of the Tibetan ice cores

(Thompson et al., 2000), Chinese Loess Plateau data (An, 2000; Porter, 2001) and stalagmites from the Hulu Cave (Wang et al., 2001) and the Dongge Cave (Dykoski et al., 2005). Paleoclimatic reconstructions reveal significant changes in Asian monsoon intensity during the MH. During the MH, the Indian monsoon penetrated deeper into the Indian subcontinent (Wang et al., 2010; Herzschuh, 2006; Kutzbach and Street-Perrott, 1985), although detailed knowledge is still limited by the absence of high-resolution proxy information. Furthermore, owing to a

lack of paleoclimate proxies from SEA, little is known about the incursion and variability of the SEAM.

To better understand orbital-scale monsoon changes, various numerical experiments have been performed, primarily using global climate models (GCMs) to simulate the large-scale atmospheric circulation and the related monsoon circulation (Jalihal et al., 2019a; Jiang et al., 2013; Dallmeyer et al., 2013). The earliest modeling studies of the MH monsoons focused only on the direct effect of insolation forcing (Kutzbach and Otto-Bliesner, 1982;

Kutzbach and Guetter, 1986; Kutzbach and Gallimore, 1988). These studies generally produced strengthened monsoons and increased precipitation in Asia in response to the increased land–sea surface temperature contrast in summer caused by the enhanced seasonal cycle of MH insolation. Although the early GCM simulations were based on prescribed SSTs and land vegetation, recent studies have shown that land and ocean feedbacks can enhance the orbitally-forced monsoon changes (Zhao and Harrison, 2012; Kutzbach et al., 2001). However, due

to their coarse horizontal resolutions, GCMs are incapable of realistically capturing local-scale atmospheric circulation and precipitation processes, which are strongly influenced by the orography. In order to fill the need for high quality climate information on regional scales while maintaining the computational tractability of the problem, this study employs the same dynamical downscaling pipeline described in Huo and Peltier (2021), in which a regional climate model—the Weather Research and Forecasting (WRF) Model is coupled to a regional

ocean model—the 3D Coastal and Regional Ocean Community model (CROCO) and both are forced with the output from a GCM—the University of Toronto version of the NCAR Coupled Climate System Model version 4 (UofT-CCSM4)—at the lateral boundaries (Fig. 2). This pipeline has been applied in the projection of modern



global warming impacts for SEAM and results verified to be in good agreement in the general monsoon patterns compared to instrumental era observations (Huo and Peltier, 2021). Similar simulations have also been carried

out for the Indian monsoon region for present-day and future climate projections by Huo and Peltier (2019, 2020) with a somewhat simplified pipeline, that didn't include the CROCO regional ocean model and utilized an older version of WRF (V3.9.1).

The MH period was also characterized by one of the greatest climate differences compared to the preindustrial (PI) and present-day climate—namely the existence of a Green Sahara (GS), wherein northern Africa was

considerably wetter than today and was covered to a great extent by a mixture of shrubland, grassland, trees, and wetlands (Pausata et al., 2020; Holmes and Hoelzmann, 2017; Chandan and Peltier, 2020). While it is well established that the Sahara existed in a state very different from that of today during the MH (Hély et al., 2014; Harrison and Bartlein, 2012) and had far-reaching local and global climatic influences (Chandan and Peltier, 2020; Pausata et al., 2017a, 2017b), most studies investigating the Asian monsoon variability during the MH have

employed the same global vegetation distribution as that characteristic of the PI, and have been demonstrated to significantly underestimate the strength and extent of the SAM and SEAM (Zhao and Harrison, 2012; Braconnot et al., 2012). Since other key factors, including remote changes in vegetation cover have been shown to modify the climate response to the insolation forcing (Piao et al., 2020; Sun et al., 2019), we have taken care to incorporate GS boundary conditions (Chandan and Peltier, 2020) in the analyses to be described in the current paper in order

to compare with the standard MH simulations and to capture interactions between the African Humid Period Monsoon and that in India and SEA. Griffiths et al. (2020) have suggested that the transition out of the African humid period led to an amplification of mid- to late Holocene megadroughts in mainland Southeast Asia via ocean-atmospheric teleconnections. Moreover, some recent studies have projected future increases in Sahelian precipitation, as well as a surface greening in the Sahel (Giannini and Kaplan, 2019; Evan et al., 2016). Hence,

understanding the influence of African vegetation on the SAM and SEAM may also help us better appreciate the changes in the Asian monsoon system to be expected from ongoing climate change and human activity.

This paper presents an analysis of the MH monsoon using high-resolution coupled RCM simulations. The key questions are: (1) How do SAM and SEAM precipitation respond to the MH orbital forcing on regional scales? (2) Do simulations produced by the UofT-CCSM4-WRF-CROCO coupled modeling system agree both

qualitatively and quantitatively with proxy data during the monsoon season? (3) What are the impacts of the GS boundary conditions on SAM and SEAM? The coupled model system and the simulation configurations are





described in Sect. 2. A synthesis of major monsoon changes is provided in Sect. 3, including a discussion of the sensitivity of the SAM and SEAM to MH Sahara vegetation changes. Section 4 offers a summary of our conclusions.

## 2 Model descriptions and experimental design

The dynamical downscaling methodology employed for the analyses to follow is the same as that in Huo and Peltier (2021), which further develops the dynamical downscaling "pipeline" introduced in Gula and Peltier (2012) and then applied in d'Orgeville et al. (2014), Erler and Peltier (2016) and Peltier et al. (2018) for the purpose of investigating the expected precipitation changes over the Great Lakes Basin of North America and over western Canada, upstream and downstream of the Rocky Mountains topographic barrier. The regional atmospheric model component of the pipeline is the WRF model Version 4.1 (Skamarock et al., 2008), with the same choice of major physics parameterizations as in Huo and Peltier (2021) except for the cloud microphysics parameterization. Unlike Huo and Peltier (2021), who employed two different microphysics schemes, this study uses a single microphysics parameterization scheme (the single-moment 6-class scheme; Hong and Lim, 2006) for all the experiments because the choice of the microphysics scheme was found to have much smaller impact on the simulated monsoon precipitation compared to the cumulus scheme in the modern day simulations of both SAM (Huo and Peltier, 2020) and SEAM (Huo and Peltier, 2021). The same four cumulus parameterization schemes as in Huo and Peltier (2021), namely the Tiedtke scheme (Tiedtke, 1989), the Grell–Freitas ensemble scheme (GF; Grell and Freitas, 2014), the Betts–Miller–Janjic scheme (BMJ; Janjic, 1994) and the Kain-Fritsch scheme (KF; Kain, 2004) are employed in this study to construct a mini-physics ensemble (Table 1) for the SAM and the SEAM, which enables us to study the sensitivity of model performance to different cumulus parameterization schemes and thereby to estimate the uncertainty associated with these parameterizations on the simulated MH climate. In the radiation module of WRF, we have applied trace gases and orbital parameters based on the Paleoclimate Modeling Intercomparison Project Phase 4 (PMIP4; Otto-Bliesner et al., 2017) protocol for the MH simulations. The WRF domain covers most of the Asian landmass (shaded in Fig. 3) at 30 km resolution.

The 3D CROCO ocean model has also been coupled to the WRF domain using the OASIS 3 coupler (Valcke, 2013) through which hourly prognostic fields are exchanged (Fig. 2). This study employs the same ocean domain as in Huo and Peltier (2021), which covers most of the oceanic area in the WRF domain including the Bay of Bengal (BOB), the eastern part of the Arabian Sea and the western part of the South China Sea (the white



rectangular area in Fig. 3) and operates at a horizontal resolution of 0.2 ° (approximately 20 km) with 32 vertical levels.

Both WRF and CROCO are forced at their lateral boundaries by the atmospheric and oceanic fields obtained from simulations with a global model. The global simulations have been generated using a fully coupled atmosphere-ocean-land-sea-ice GCM: the University of Toronto version of the National Center for Atmospheric Research

(NCAR) Coupled Climate System Model version 4 (UofT-CCSM4; Peltier and Vettoretti, 2014; Chandan and Peltier, 2017, 2018, 2020). The UofT-CCSM4 simulations were produced in its coupled configuration and with a horizontal resolution of 1°. More details regarding these simulations can be found in Chandan and Peltier (2020). For the analyses reported herein, WRF-CROCO was driven by three distinct GCM simulations, including one for the PI climate, and two for the MH climate, the latter being forced by the PMIP4 mandated concentrations of

greenhouse gases (GHGs) and orbital parameters (Otto-Bliesner et al., 2017). One of the MH experiments, henceforth referred to as the reference MH simulation and denoted as $MH_{REF}$, employs a preindustrial land surface and therefore differs from the PI only with regards to the orbital configuration and trace gas concentrations; the other MH experiment, henceforth referred to as $MH_{GS}$, also incorporates GS boundary conditions by including vegetation, soil and lake modifications over Africa and Arabia. Both of these simulations were recently described

in detail in Chandan and Peltier (2020), where they were referred to as $MH_{REF}$ and $MH_{VSL}$ respectively. Each Uoft-CCSM4 experiment was integrated for several hundred years and the last 15 years of output of each were downscaled using WRF-CROCO for the purpose of the analyses to be reported below. The two sets of MH experiments, both including results for each member of the four member mini-ensemble, can be compared to investigate the role of the Saharan landscape in altering the SAM and SEAM. We will also compare the simulated

changes in precipitation in the two sets of experiments to pollen-based climate reconstructions (Bartlein et al., 2011) to help us evaluate the degree to which the incorporation of GS boundary conditions will lead to a diminution of the previously obtained misfits of global model-based climate reconstructions of SAM and SEAM rainfall.

## 3 Results and discussion

We begin by discussing the sea surface temperature (SST) and surface air temperature anomalies simulated for $MH_{REF}$ before turning our attention to precipitation changes realized in the same simulation. Following this, we discuss the impact of including the full GS boundary conditions on the simulated MH SAM and SEAM. All





spatially-averaged anomalies reported here are calculated over the Indian subcontinent or mainland SEA south of the TP (the two black rectangles in Fig. 3). These two analysis regions are identical to the inner WRF domains in Huo and Peltier (2020, 2021) wherein two levels of downscaling were employed.

**3.1 Ocean SST and surface atmospheric temperature changes in MH$_{REF}$**

The establishment of the Asian monsoon circulations is determined by the seasonal cycle of SSTs of the Indian Ocean and the contrast with the surface temperature variation over the land mass to the north and east. Seasonal variations in the strength of the monsoon are therefore closely connected with variations in SSTs (Clark et al., 2000). Figure 4 shows the SST anomalies, with respect to the preindustrial, simulated for MH$_{REF}$ by the GCM and by the first WRF-CROCO ensemble member (Table 1). The simulated SSTs from the other three ensemble members (not shown) are not significantly different. In the downscaled MH$_{REF}$ simulation, SSTs in JJAS (June-July-August-September) are 0.5° C colder than in the PI in the northeastern Arabian Sea (Fig. 4b), whereas the global model produces larger negative anomalies (up to -0.8° C) across the eastern Arabian Sea. In contrast, the JJAS SST anomalies of up to -1° C that are simulated for the BOB in the downscaled simulations produced with CROCO are larger than those in the global model. This extended cooling all along the eastern coastline of India is likely the result of stronger coastal upwelling driven by the strengthened MH monsoonal winds. CROCO also simulates warm SST anomalies (up to 1° C) around the southern tip of India, whereas UofT-CCSM4 simulates cold SST anomalies over the entire ocean domain.

In response to the MH changes in the orbital parameters and GHG concentrations, the 2 m air temperature over land in summer also changes significantly compared to the preindustrial (Fig. 5). During JJAS, land temperatures in the MH are warmer by up to 2° C northward of the TP and the largest warming center occurs at around 45° N. The intensity of this warming center as well as its spatial extent in the downscaled simulation (Fig. 5b) is larger than that in the GCM (Fig. 5a). The Indian subcontinent itself experiences a small cooling (approximately 0.5° C), whose expression does not differ significantly between the GCM and WRF. A summer cooling over SA of similar magnitude was also found in PMIP4 MH simulations (Brierley et al., 2020). The Indochinese Peninsula features small temperature anomalies of opposite signs in the western and eastern regions, leading to no significant average temperature change over the peninsula (Fig. 5b). As a result of these changes, the MH summer land meridional temperature gradient decreases through the low to mid-latitudes. At the same time, the land–ocean thermal contrast increases due to the rise in the spatially averaged JJAS temperature over the continent and decrease in temperature over the sea, which intensified the moisture transport towards the continent within the





low-level monsoonal wind field (see sect. 3.4). Moreover, the MH forcing yields a climate that is colder over SA and SEA during three quarters of the year (Fig. 5c) and warmer only from midsummer to early fall.

### 3.2 Average precipitation changes in MH$_{REF}$

These changes in the SST and the temperature distribution over land and sea have a large impact on the summer monsoon wind field, moisture transport towards the Eurasian landmass, as well as the regional precipitation distribution. Figure 6 shows the annual mean precipitation difference between the MH$_{REF}$ and the PI and its comparison to proxy-inferred precipitation changes. Under modern climate conditions, the SAM has highest precipitation rates over the west coast of the Indian subcontinent associated with the Western Ghats mountains

and in the north along the Himalayan foothills (Fig. 4 in Huo and Peltier, 2020), whereas the SEAM produces the most intense precipitation over the west coast of Myanmar (Fig. 5 in Huo and Peltier, 2021). For MH$_{REF}$, most of SA experiences a wetter climate compared to the PI, which is consistent with the results obtained in other PMIP3 and PMIP4 model experiments (e.g. Zhao and Harrison, 2012; Brierley et al., 2020). Annual mean rainfall is increased by up to 2 mm day$^{-1}$ in northeastern India and along the windward side of the Western Ghats foothills

and by approximately 1 mm day$^{-1}$ in the northern part of SEA in WRF-CROCO simulations (Fig. 6c). These increases are much weaker in the global UofT-CCSM4 model, while the locations of increased precipitation along the Himalayan Range differs from WRF-CROCO: in the global model the largest precipitation increase occurs over the Hindu Kush and Pamir mountain ranges in the western Himalayas, whereas in the downscaled model precipitation increases over a broad region of the Himalayan foothills extending from Nepal to the India-China-

Myanmar border in the east. The global model on the other hand simulates drying over the eastern Himalayas. These substantial differences between the global and the regional model results attest to the importance of high resolution modeling over regions of complex land surface. Over SEA, decreased precipitation (approximately 0.5 mm day$^{-1}$) is delivered by both global and regional models over the southern region, which is consistent with the results obtained in other PMIP4 simulations (Brierley et al., 2020). Over Southern India, the global model

produces a decrease in annual precipitation while the WRF-CROCO simulation shows an increase. This is likely related to the differences in the simulated SST anomalies in these simulations around the southern tip of India, where UofT-CCSM4 produces a negative anomaly while the higher-resolution ocean model shows warmer SSTs in MH (Fig. 4).

       Figure 6 also compares the simulated annual precipitation anomalies with the pollen-based paleoclimatic

reconstructions (Bartlein et al., 2011) in order to evaluate the degree to which the dynamical downscaling-based





analysis is able to improve the simulation of MH monsoon rainfall. Overall, there is qualitative agreement between the palynology-based reconstructions and the model predictions with wetter conditions in the northern region of SEA and along the windward slopes of the western and central Himalayas. However, the model MH prediction disagrees with the reconstructions in the eastern part of the TP, where the model produces a negative precipitation
anomaly while the reconstructions indicate increased rainfall in the MH. Overall, the downscaled model simulation displays a slightly greater similarity in pattern to reconstructions than the GCM in the northern part of SEA and south-western China, although they both continue to significantly underestimate the inferred changes in precipitation. Both the GCM and downscaled simulations generally suggest a drier climate in central and eastern China during the MH, while the pollen-based reconstructions suggest predominantly wetter conditions. However,
even the proxy-based reconstructions are somewhat contradictory from site to site, which may result from topographic differences or local hydrological processes. Additionally, stalagmite records from seven caves in Asia (Cai et al., 2012; Dykoski et al., 2005; Cai et al., 2010; Cosford et al., 2008; Hu et al., 2008; Dong et al., 2010; Zhang et al., 2004) all indicate generally high monsoon precipitation during the MH. The locations of the seven caves (Tianmen, Dongge, Jiuxian, Lianhua, Heshang, Sanbao and Xiangshui Caves) are shown as "*" in Figs. 6b
and 6c. All six stalagmite records over East China are qualitatively in accord with results from the coupled WRF-CROCO simulation, while the GCM simulates a wetter climate at the locations of five of the caves but a drier climate at the location of Xiangshui Cave (25.3° N, 110.9° E). At the location of the Tianmen record over the TP, UofT-CCSM4 simulates no significant change while the downscaled simulation indicates a slightly drier climate during the MH. Since regional precipitation is characterized by high spatial variability and reconstructions
inherently have large uncertainties, more proxy data would be needed for a more comprehensive comparison between the model simulations and the reconstructions, particularly over India and SEA.

Annual precipitation is strongly controlled by summer rainfall, so the JJAS precipitation anomaly patterns (Fig. 7) resemble those of the annual mean. However, compared with the distribution of annual rainfall changes (Fig. 6), monsoon rainfall increase is greater (up to 3 mm day$^{-1}$) during the monsoon season over northern SA and SEA.
The intensification of the monsoon results in increased cloudiness and evaporation, thereby contributing to the lower temperature over northern SA and SEA (Fig. 5).

Comparing different ensemble members, all physics configurations provide generally consistent signals for both SA and SEA during JJAS, with greater monsoon precipitation increase over northern SA and SEA and along the windward slope of the Western Ghats (Fig. 7). Among all the ensemble members, the GF scheme in the second
ensemble member produces the largest JJAS precipitation increase (1.4 mm day$^{-1}$), while the first ensemble





member based on the KF cumulus scheme produces the smallest wet anomaly over SEA (0.3 mm day$^{-1}$), which is a result of the large dry anomaly in the south compensating the rainfall increase in the north (Fig. 7d). Over SA, however, the KF cumulus scheme produces the largest wet anomaly among all ensemble members (1.3 mm day$^{-1}$), while the third ensemble member using the Tiedtke cumulus scheme has the smallest precipitation change (0.8 mm day$^{-1}$). Furthermore, the GF scheme produces the second largest JJAS rainfall increase over SA (1.2 mm day$^{-1}$). Such different JJAS rainfall signals over SA and SEA indicate the impact of the choice of different cumulus schemes on the spatial distribution of moisture. In our simulations of SA and SEA for the present day (Huo and Peltier, 2020, 2021), the KF cumulus scheme performed the best in terms of reproducing the mean SAM precipitation of all cumulus schemes, while it significantly overestimated precipitation over nearly all of SEA. The Tiedtke scheme has a general dry bias over SA, while it can be considered the best in terms of the SEAM average precipitation simulation during instrumental era. The BMJ cumulus scheme generally produces a moist bias in both the Indian and SEA cases under modern climate conditions.

Based on the annual cycle of precipitation over SEA (Figs. 6d and 7f), monsoon intensity is actually weaker during the early stage (June) in both the global model and all WRF-CROCO ensemble members, although the decrease is much greater in downscaled simulations than in the GCM. This rainfall reduction during the early stage of the monsoon indicates that SEAM starts later in the MH than in PI, which is consistent with previous studies (Dallmeyer, 2013). The late monsoon onset is likely related to the lower temperature on the Indochina peninsula during spring and early summer (Fig. 5e). On the other hand, the maximum precipitation increase over SEA occurs in September, which is near the end of the Asian monsoon season. Furthermore, the precipitation increase in October suggests that the withdrawal of the summer monsoon is postponed in Indochina. Over SA, the peak monthly rainfall increase occurs in August and September in all ensemble members, and October experiences a very small increase according to three of the ensemble members, with the ensemble member employing the GF cumulus scheme being the exception. Although the monthly rainfall changes due to MH forcing are more drastic over SEA, the overall monsoon (JJAS) rainfall increase is actually smaller over SEA than SA due to the large dry anomaly in June that compensates the increased precipitation during late summer. Furthermore, all of these monthly monsoon rainfall changes discussed above are stronger in the downscaled simulations than in the GCM (Figs. 6b, 6c, 7e and 7f). The spatially-averaged JJAS precipitation increases produced by UofT-CCCSM4 are only 0.5 mm day$^{-1}$ over SA and 0.2 mm day$^{-1}$ over SEA.



### 3.3 Impact of the GS boundary conditions

To investigate the land-ocean-atmosphere feedbacks and teleconnections between GS boundary conditions, that existed over northern Africa and Asian monsoons, another four-member physics ensemble, forced with global simulations that included GS boundary conditions was constructed.

In the MH$_{GS}$ simulations, the influence of GS surface boundary conditions (Chandan and Peltier, 2020) leads to JJAS temperature anomaly over Asia that has a lot in common with MH$_{REF}$, with higher latitudes serving as the

primary center of warming (Fig. 5b and Fig. 8a). However, except over India, the warming in MH$_{GS}$ is more pronounced than that in MH$_{REF}$ over the majority of the WRF domain, with the largest increase over the TP and East Asia (Fig. 8b). Such enhanced warming is conducive to increasing the northward moisture flux to SA and SEA by increasing the strength of the low-pressure center over the continental interior. JJAS temperature over India is cooler than in the experiment without GS forcing, which may be linked to the further increase in albedo

resulting from the greater increase in cloud fraction and JJAS rainfall over the region (Fig. 9) and is similar to previous modeling results (Pausata et al., 2020). Over SEA, inclusion of a GS leads to increased warming year round with the largest increase in January (up to 0.9° C, Fig. 8c). With a GS the climate becomes warmer than present over most of SEA by approximately 0.3° C during JJAS (Fig. 8a), whereas the MH orbital and trace gas forcing alone (MH$_{REF}$) produces a small cooling over western SEA (Fig. 5b). Compared to MH$_{REF}$ the SSTs over

the Indian ocean and the South China Sea are now warmer everywhere by at least 0.5° C, with pockets of even warmer SSTs to the south of India and Vietnam. As a result of this warming, the warm SST center south of India that was present in MH$_{REF}$ intensifies in strength (up to 2° C warmer than PI, Fig. S1) while the cooling over the northeastern Arabian Sea and southern BOB is reduced.

The inclusion of a GS significantly reinforces the precipitation anomaly pattern that was simulated over SA only

in the presence of MH orbital insolation and GHG forcings (Fig. 9c). The positive rainfall anomaly is particularly large in northeastern India and along the Western Ghats (up to 3 mm day$^{-1}$). In south India, whereas there were no significant precipitation changes in the rain shadow east of the Western Ghats in MH$_{REF}$ (Fig. 6c), there is now a strong increase in precipitation in the simulations with GS forcing (Fig. 9c). Over SEA, feedbacks from land surface changes over northern Africa also lead to a precipitation intensification relative to MH$_{REF}$, albeit with a

smaller amplitude than that over SA (Fig. 9c). The positive rainfall anomaly in the northern region of the Indochina peninsula extends southward, reducing the negative precipitation anomaly that was modeled over the southern region in MH$_{REF}$. The decrease in precipitation over the northern BOB in MH$_{REF}$ is also reduced.



Generally, when the north African land surface changes in the MH are included, the modeled annual mean precipitation change over most of the Asian monsoon regions shows a significantly improved agreement with

proxy records, particularly over the northern and western TP and over northeastern China (Fig. 9), compared to that which is obtained only in the presence of MH orbital and GHG forcings (Fig. 6). Comparing $MH_{REF}$ and $MH_{GS}$ from the UofT-CCSM4 (Figs. 6a and 9a), inclusion of a vegetated Sahara during the MH enhances precipitation along the Himalayas and produces a positive anomaly band stretching from the western TP to northeastern China, which leads to an improvement of model-proxy agreement and is consistent with Tabor et al.

(2020). With dynamical downscaling, the model better captures both the sign and magnitude of the local-scale response characteristics of the MH proxy records (Figs. 9a and 9b), especially the precipitation enhancement along the Tian Shan north of the TP and from the northern SEA to North China, thus further improving the proxy-data comparison. However, the precipitation increase over the eastern TP and southeastern China that is revealed by proxy data is still not captured by dynamical downscaling or by incorporating the influence of the greening of the

Sahara.

The JJAS precipitation anomalies in the experiments with GS boundary conditions are generally consistent among different WRF-CROCO ensemble members: both the SAM and SEAM rainfall significantly increase compared to the PI climate as well as to $MH_{REF}$ (Fig. 10). All ensemble members are characterized by increased wet anomalies in the north than in the south in both SA and SEA. In terms of the spatially averaged JJAS precipitation

over SA, in almost all ensemble members the presence of GS conditions leads to a doubling of the precipitation increase produced by $MH_{REF}$. On the other hand, over SEA, the amplitude of the increase in $MH_{GS}$ is less consistent between the ensemble members. The first ensemble member, which is characterized by small overall JJAS rainfall changes over SEA in $MH_{REF}$ due to the opposite signs of anomalies simulated in the south and the north, triples the wet anomaly in $MH_{GS}$ by reducing the widespread drying over southern SEA. In the fourth

ensemble member using the BMJ cumulus scheme, feedbacks from the Sahara greening leads to the smallest precipitation intensification compared to the simulation with only the orbital/GHG forcing (0.3 mm day$^{-1}$) among all ensemble members, which is only 30% of the averaged wet anomaly in $MH_{REF}$ (0.9 mm day$^{-1}$).

While JJAS average rainfall determines the overall amount of water supply, shifts in monsoon onset and withdrawal play an important role in determining the length and thus the net precipitation that falls during the

monsoon season. In the PI experiment, the SAM starts in late May, and then develops until August and retreats in September (Fig. 11a). In $MH_{REF}$, the monsoon onset is delayed and the precipitation reduction is greater at lower



latitudes (Fig. 11b). The impact of the late onset is offset by a delayed withdrawal, which lengthens the SAM duration. Inclusion of the GS boundary conditions results in a further lengthening of the monsoon season to November (Fig. 11c). The delay of the monsoon onset in MH$_{REF}$ is also eliminated by including a vegetated Sahara, although the region at lower latitudes is still characterized by a relatively smaller precipitation increase in June. On the contrary, SEA has a much stronger delay of the monsoon onset and shows a greater decrease in rainfall in early summer in MH$_{REF}$ (Fig. 11e). The withdrawal of the SEAM is also delayed by about one month. MH$_{GS}$ displays a weaker decrease in precipitation than MH$_{REF}$ in spring and early summer. The Sahara greening also increases the rainfall at the end of the monsoon season (September and October) especially at the higher latitudes, further postponing the monsoon withdrawal.

### 3.4 Changes in the large-scale moisture flux

The JJAS precipitation over SA and SEA is closely related to the summer monsoonal winds transporting moisture from the ocean to the Eurasian continent (Fig. 12d). The enhanced Northern Hemisphere insolation during the MH and enhanced land-sea temperature gradients (Fig. 5) results in an intensification of the westward flow around the southern tip of India and the eastward wind across SA and SEA, intensifying the simulated Asian monsoonal circulation and strengthening the moisture flux to SA and SEA (Fig. 12e). This spatial pattern of wind anomaly agrees well with previous studies (Pausata et al., 2020; Piao et al., 2020) and explains the rainfall changes in Fig. 6. Furthermore, the negative SST anomalies over the northern Arabian Sea probably results from the enhanced upwelling caused by the intensified monsoonal flow and the increased SSTs south of the Indian subcontinent are likely related to the weakened westerlies over that region (Fig. 4). The convergence of the simulated low-level moisture flux has a positive anomaly over northern SA and South China, while the water vapor flux becomes slightly more divergent over southern SEA and southern and central SA in MH$_{REF}$. As a consequence, the enhanced moisture flux and convergence over northern India lead to a greater JJAS precipitation increase than in southern India during the MH. Conversely, the reduction of JJAS precipitation in the southern part of SEA is connected to the decreased moisture flux convergence in that region. In mid-latitudes, the high-level moisture flux above the Eurasian continent becomes more divergent in MH$_{REF}$ than in the PI (Fig. 12b). Under the GS boundary conditions, the low-level westerlies over the BOB becomes slightly weaker than in MH$_{REF}$ (Fig. 12f), which decreases the upwelling and hence increases the SSTs of that region (Fig. S1). Although the south-westerlies over the BOB weaken, the moisture divergence over the southern part of the Indochinese Peninsula is reduced (Fig. 12f), which explains the smaller precipitation decrease of that region (Figs. 6c and 9c). Over the rest of the Asian



monsoon domain, the inclusion of a vegetated Sahara intensifies the anomaly pattern characteristic of the MH$_{REF}$ experiment, in particular over India, which explains the strong precipitation increase over most of SA (Fig. 9).

## 4 Conclusions

In this study, we have reported on paleoclimate experiments with dynamical downscaling simulations that we
have conducted using the coupled WRF-CROCO model driven by the output from the global coupled model UofT-CCSM4 GCM, to investigate the MH Asian regional climate. We have focused attention on the summer monsoon circulations of this region, reconstructing these at much higher resolution than is possible with a global model by employing a dynamical downscaling procedure. By employing a downscaling pipeline in which the reginal climate model WRF is coupled to the regional ocean model CROCO we have been able not-only to capture
the effects of the complex surface orography and land-sea contrast of the region but also achieve a much higher-fidelity data-model comparison than has been achieved previously. Furthermore, and in contrast to the common protocol for climate simulations of the MH Asian monsoon system (Otto-Bliesner et al., 2016), which only includes the impact of solar insolation changes and ignores the potential for teleconnection effects on the Asian monsoons, our study also incorporates a set of downscaled simulations that include Green Sahara (GS) boundary
conditions through which we have been able to assess the remote response of the MH SAM and SEAM systems to the greening of the Sahara. These effects have been shown to be of first order in importance. The magnitude of the impacts are significantly enhanced by the high spatial resolution delivered by the downscaling pipeline.

Compared to the pre-industrial, all physics ensemble members, downscaled from the reference MH climate which does not include a GS, consistently exhibit a strengthened summer monsoon intensity during the MH over both
SA and SEA. The increase in summer rainfall is particularly strong over northern India and northern Indochina, while monsoon precipitation decreases over the southern part of SEA and the BOB. MH monsoon intensity is weaker during early summer, particularly over SEA, which is likely linked to the lower SSTs in the east Arabian Sea and the BOB. A late monsoon onset and a delay in the monsoon withdrawal over both SA and SEA are also found in our analysis. All of these changes are stronger in the downscaled simulations than in the GCM.

Comparison with paleoclimatic reconstructions from the Bartlein et al. (2011) demonstrates that over northern Indochina and along the Himalayas, the simulated annual rainfall anomalies agree well with the reconstructions from a qualitative perspective. However, over the eastern TP, the model simulations are characterized by a drier climate while wetter conditions are indicated by proxy data for the MH. Overall the WRF results are in better



agreement with proxy inferences especially in the northern part of SEA and the south-western China than the predictions of the UofT-CCSM4 global model which is much coarser in resolution and does not resolve the regional orography accurately. Given that climate models using coarse spatial resolutions generally reproduce the direction and large-scale patterns of the MH monsoon system changes, but tend to underestimate the regional response (Hargreaves et al., 2013; Zhao and Harrison, 2012; Braconnot et al., 2012), this study clearly demonstrates the added value of using dynamical downscaling in paleoclimate studies of the Asian monsoon system. However, since proxy data are scarce both geographically and temporally, particularly over SEA and the southern part of SA, more reconstruction work based on various proxies and methods will be required to draw more robust conclusions.

Among the individual physics ensemble members, there remains a non-trivial degree of uncertainty in the simulated changes to the magnitude and spatial distribution of the MH SEAM and SAM. The first ensemble member based on the KF cumulus scheme produces the smallest wet anomaly over SEA (0.3 mm day$^{-1}$) and the largest wet anomaly over SA among all ensemble members (1.3 mm day$^{-1}$). The GF scheme in the second ensemble member, however, produces the largest JJAS precipitation increase over SEA (1.4 mm day$^{-1}$), and the Tiedtke scheme applied in the third ensemble member has the smallest precipitation change over SA (0.8 mm day$^{-1}$).

Comparison between simulations with and without GS boundary conditions demonstrates that greening of the Sahara amplifies the response of SAM and SEAM and leads to a further increase in MH monsoon precipitation, which is likely related to increased heating over the interior of the Eurasian continent. Over SA under GS conditions, the water vapor flux from the Arabian Sea to the northern Indian subcontinent is enhanced compared to the case when only the orbital forcing is considered. Over the southern part of SEA, although a Green Sahara weakens the low-level westerly moisture flux, it also reduces the moisture divergence. With a GS, the JJAS rainfall increases are more than twice as large as those caused by the orbital and GHG forcings in all WRF-CROCO physics ensemble members over SA. Compared with MH$_{REF}$, the KF cumulus parameterization scheme exhibits the largest JJAS precipitation increase over SEA among all the ensemble members from including the Saharan vegetation, while the fourth ensemble based on the BMJ cumulus scheme has the smallest JJAS rainfall enhancement (30%) in MH$_{GS}$. The duration of monsoon season is also lengthened by several months over both regions by extending its withdrawal phase. Comparison of downscaled results including a GS with paleoclimate reconstructions points to significant improvements in simulating precipitation, especially over the north and west sides of TP and in northeastern China, when the influence of a vegetated Sahara is taken into account. These

results highlight the climate sensitivity of Saharan vegetation changes via ocean-atmosphere teleconnections and
emphasized the importance of incorporating MH vegetation feedbacks.

Finally, in this study we have only considered the impact of land surface changes over northern Africa on SAM and SEAM using one coupled regional modeling system. However, some recent studies also argue for the impact on the MH Asian monsoon arising from a reduced dust over northern Africa (Pausata et al., 2020; Sun et al., 2019). Additionally, proxy data from the MH indicate widespread vegetation changes everywhere, particularly the
greater vegetation coverage over Eurasia (Tarasov et al., 1998) and South and East Asia (Zhang et al., 2014). It has already been shown that the higher Eurasian forest cover during the MH can shift the intertropical convergence zone northward and has remote impact on global climate (Swann et al., 2014). These results call for further investigations using both diverse numerical models and more realistic vegetation distributions and dust concentrations to enable further improvements in reconstructions of the MH Asian monsoon systems.

**Code/Data availability**

Code and Data for reproducing each of the figures in the paper can be obtained from Y. Huo.

**Author contribution**

W.R. Peltier and Y. Huo designed the experiments and Y. Huo carried them out. W.R Peltier worked to integrate CROCO into the UofT downscaling pipeline together with SciNet colleagues. D. Chandan designed
the Green Sahara simulation and performed the UofT-CCSM4 global experiments. Y. Huo adapted the WRF radiation module for the MH experiments with the help of D. Chandan. Y. Huo prepared the manuscript with contributions from both co-authors.

**Competing interests**

The authors declare that they have no conflict of interest.

**Acknowledgments**



Mr. F. Xie consulted on the CROCO coupled WRF pipeline and his inputs are also gratefully acknowledged. The simulations presented in this paper were performed at the SciNet High Performance Computing facility at the University of Toronto, which is a component of the Compute Canada HPC platform. The research of WRP at the University of Toronto is supported by NSERC Discovery Grant A9627.

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

**Table 1.** Simulations comprising the physics ensemble and their selected cumulus schemes.

| Simulation | 1 | 2 | 3 | 4 |
|---|---|---|---|---|
| Cumulus Scheme | KF | GF | Tiedtke | BMJ |





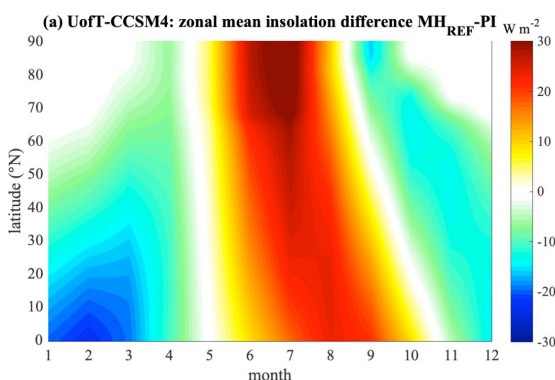

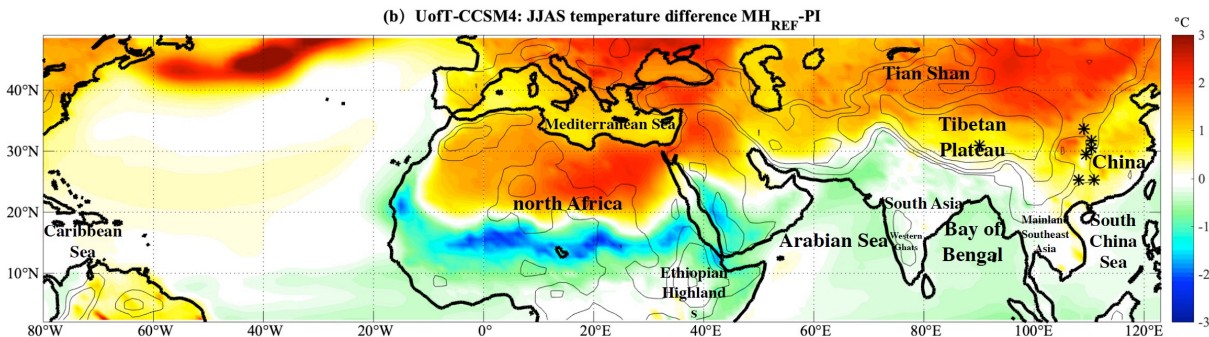

**Figure 1: a.** Zonal mean insolation difference per month between MH and PI at the top of the atmosphere in W m⁻² for MH_REF from the UofT-CCSM4; **b.** JJAS temperature difference between MH_REF and PI from the UofT-CCSM4 with the names of the geographic regions of interest. The locations of Tianmen (30.9° N, 90.1° E), Dongge (25.3° N, 108.1° E), Jiuxian (33.6° N, 109.1° E), Lianhua (109.5 ° E, 29.5° N), Heshang Cave (30.5° N, 110.4° E), Sanbao (31.7° N, 110.4° E) and Xiangshui (25.3° N, 110.9° E) Caves are indicated by "*". The topography contours of 500 m, 1000 m, 2000 m and 4000 m are also shown.



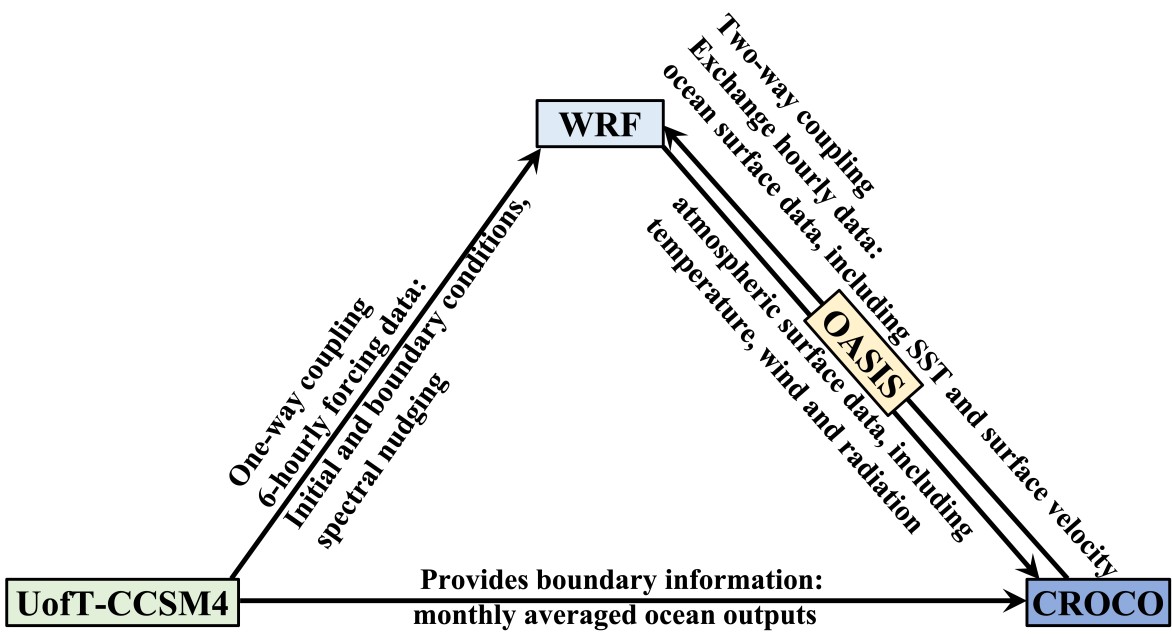

**Figure 2: Data-flow schematic of the coupling system**



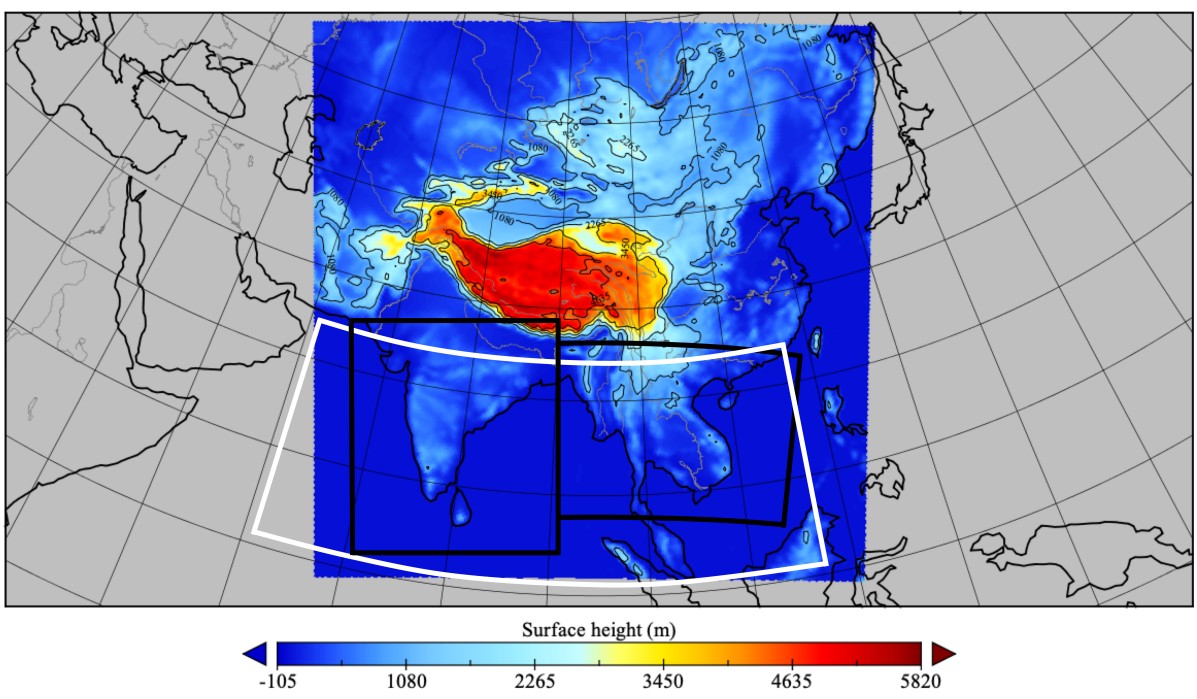

**Figure 3: Shaded topography along with the outlines of the (shaded region) WRF and (white rectangle) CROCO domains. The two black rectangles denote the regions used to calculate spatial averages over SA and SEA. Major rivers and lakes are shown in grey contours and selected topographic heights in thin black contours.**




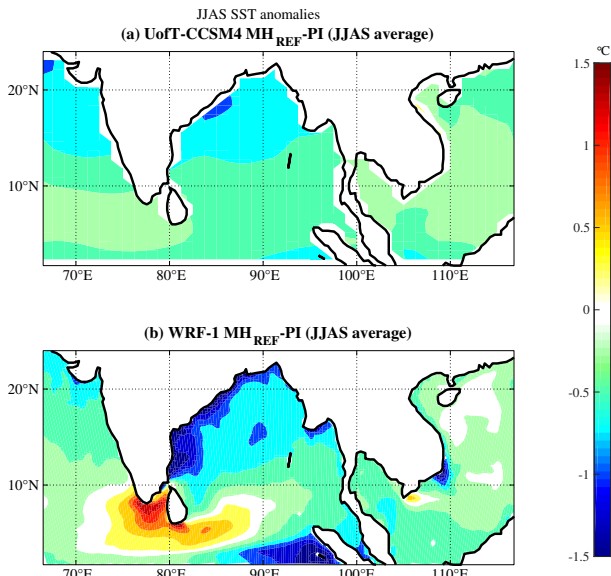

**Figure 4: JJAS SST anomalies (MH minus PI) for MH$_{REF}$ simulated by (a) UofT-CCSM4 and (b) CROCO. Shifts in calendar are not accounted for, i.e. the model calendar is used for the calculation of all anomalies.**





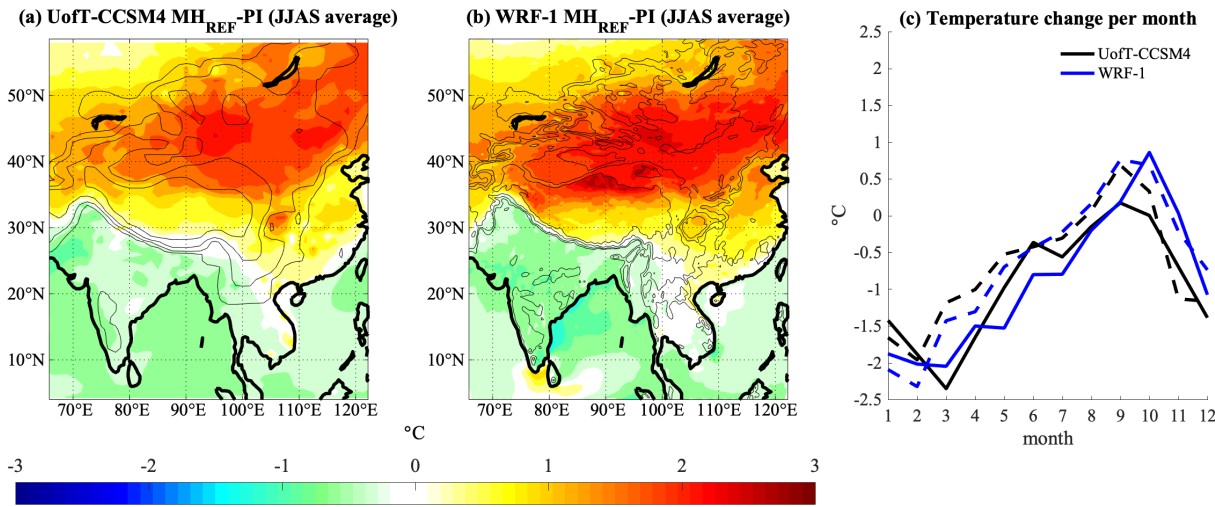

**Figure 5: JJAS surface air temperature anomalies (in ° C) for MH$_{REF}$ in (a) UofT-CCSM4 and (b) WRF. (c) shows**
**monthly temperature anomalies over SA (solid) and SEA (dashed). Shifts in calendar are not accounted for, i.e. the model calendar is used for the calculation of all anomalies. The topography contours of 500 m, 1000 m, 2000 m and 4000 m are also shown in (a, b).**



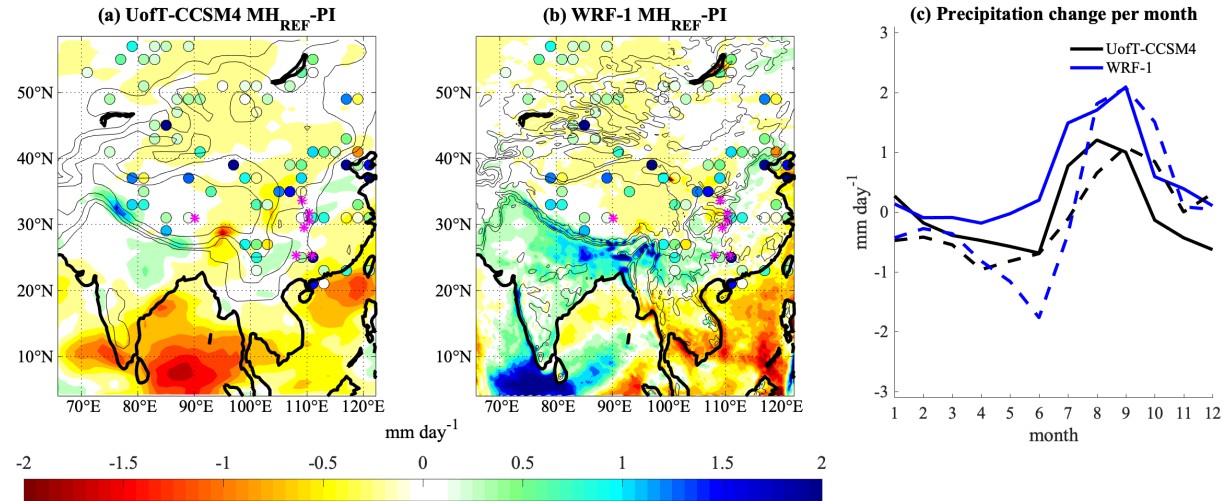

**Figure 6: (a) Reconstructed precipitation difference between MH and present-day from Bartlein et al. (2011). Annual**
**mean precipitation anomalies (mm day$^{-1}$) for MH$_{REF}$ in (b) UofT-CCSM4 and (c) WRF ensemble member 1. (d) shows**
**monthly precipitation anomalies over SA (solid) and SEA (dashed). Shifts in calendar are not accounted for, i.e. the**
**model calendar is used for the calculation of all anomalies. The locations of Tianmen (30.9° N, 90.1° E), Dongge (25.3°**
**N, 108.1° E), Jiuxian (33.6 ° N, 109.1 ° E), Lianhua (109.5 ° E, 29.5° N), Heshang Cave (30.5° N, 110.4° E), Sanbao (31.7°**
**N, 110.4° E) and Xiangshui (25.3° N, 110.9° E) Caves are indicated by "*" in (b, c). The topography contours of 500 m,**
**1000 m, 2000 m and 4000 m are also shown in (a, b).**



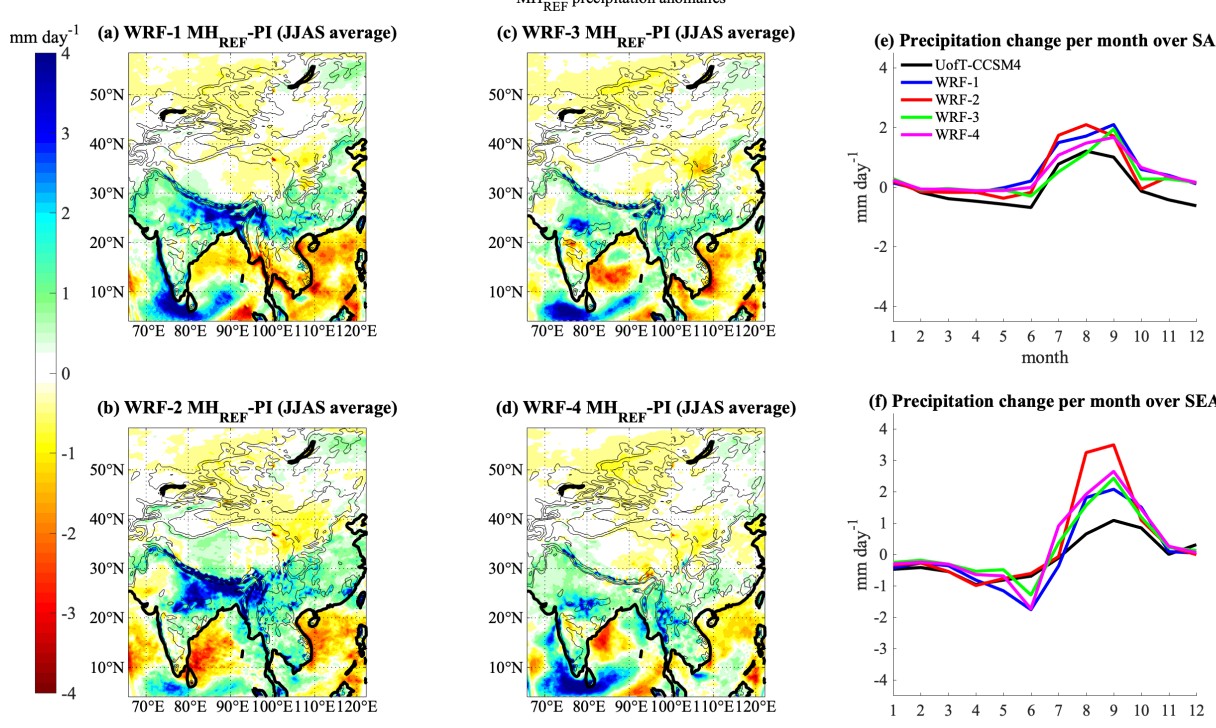

**Figure 7:** (a-d) JJAS average precipitation anomalies (mm day$^{-1}$) for MH$_{REF}$ in four WRF physics ensemble members. Monthly precipitation anomalies (mm day$^{-1}$) for MH$_{REF}$ in UofT-CCSM4 and four physics ensemble members over (e) SA and (f) SEA. Shifts in calendar are not accounted for, i.e. the model calendar is used for the calculation of all anomalies. The topography contours of 500 m, 1000 m, 2000 m and 4000 m are also shown in (a-d).






JJAS temperature anomalies

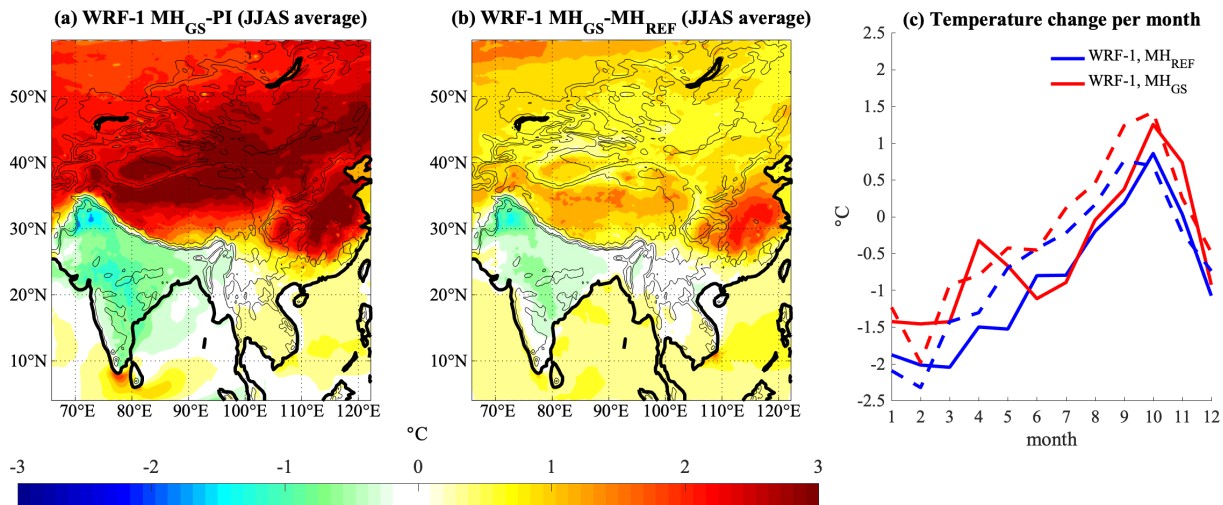

**Figure 8:** JJAS surface air temperature anomalies (in ° C) for MH$_{GS}$ (a), and differences between MH$_{GS}$ and MH$_{REF}$ (b) in WRF. (c) shows monthly temperature anomalies over SA (solid) and SEA (dashed). Shifts in calendar are not accounted for, i.e. the model calendar is used for the calculation of all anomalies. The topography contours of 500 m, 1000 m, 2000 m and 4000 m are also shown in (a, b).




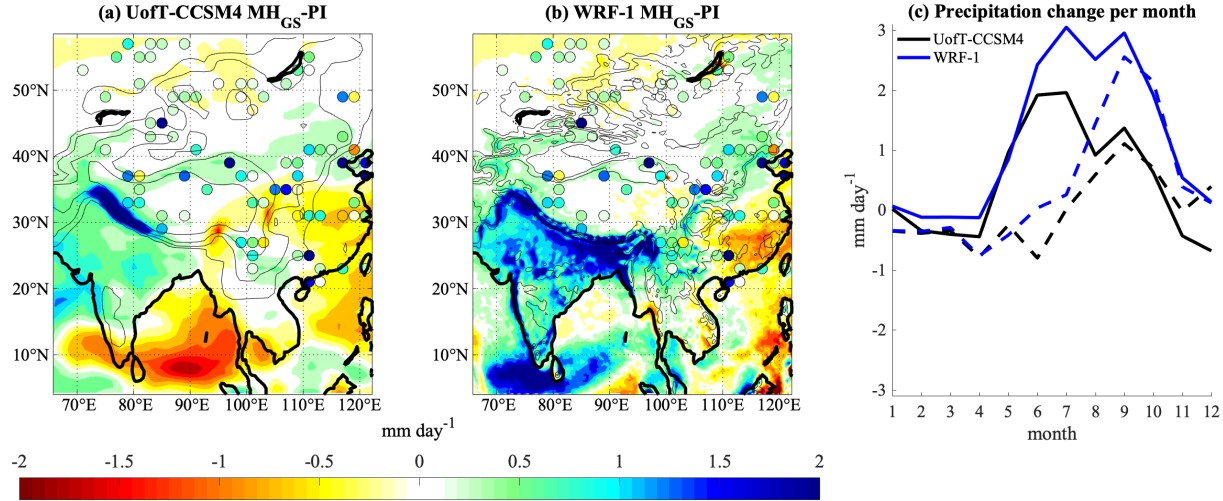

**Figure 9: Same as Fig. 6, but for MH$_{GS}$.**





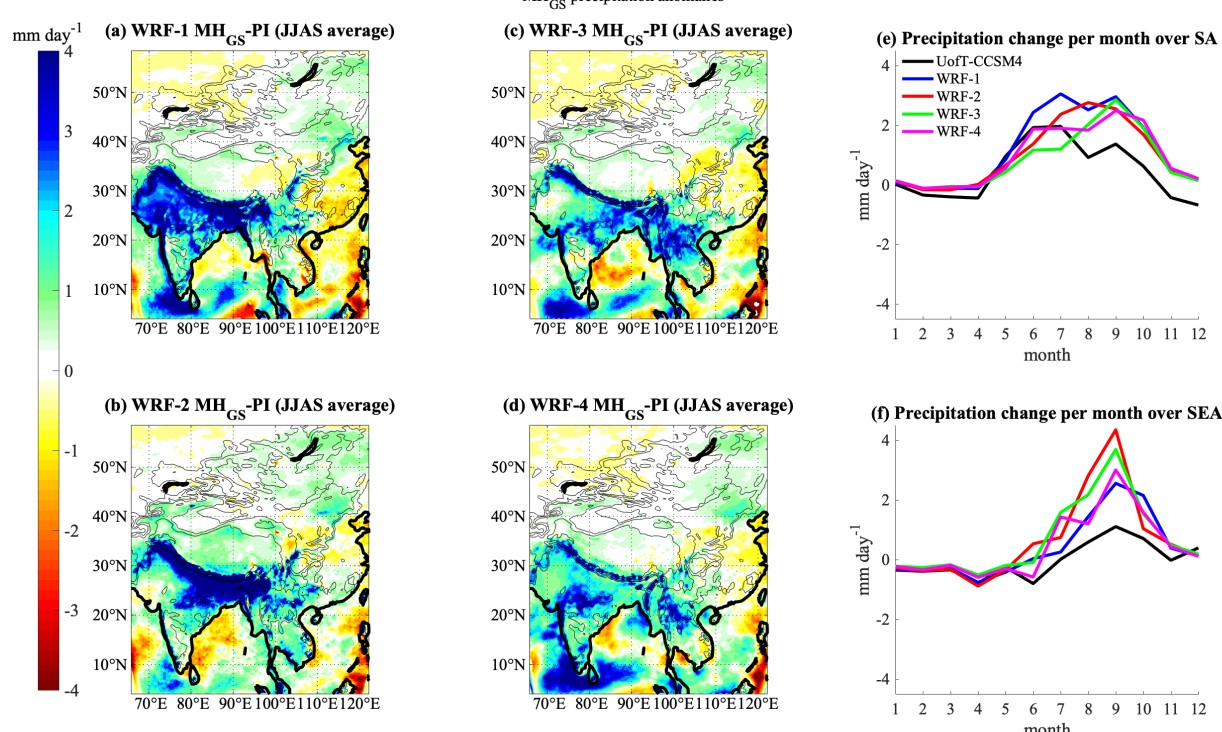

**Figure 10: Same as Fig. 7, but for MH$_{GS}$.**

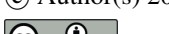



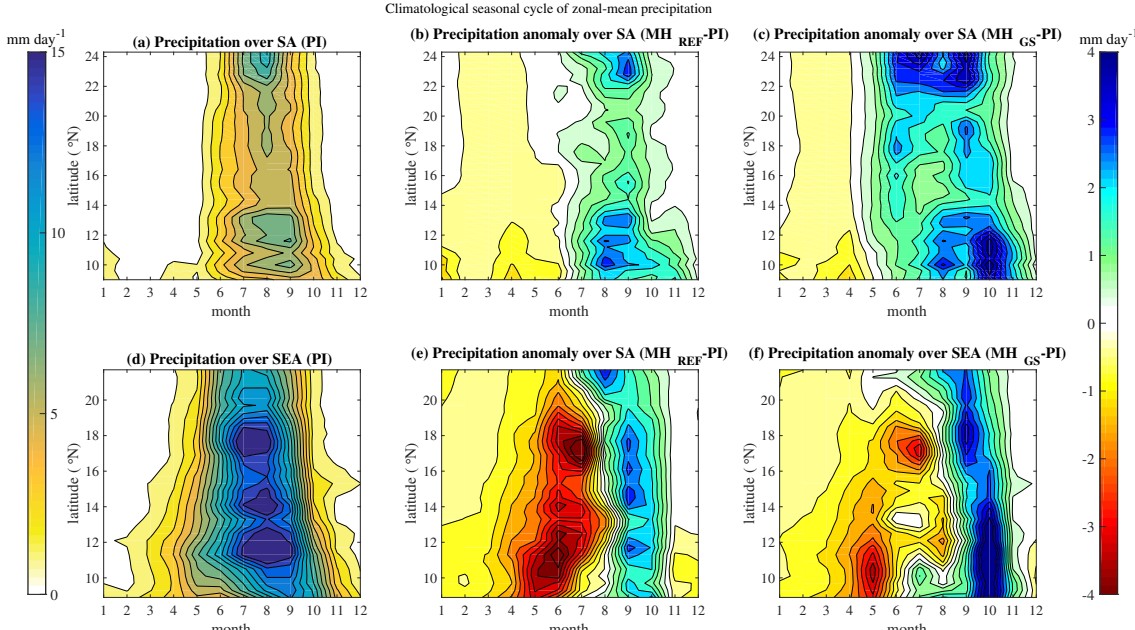

**Figure 11: Climatological seasonal cycle of zonal-mean precipitation (mm day⁻¹) for the PI simulations of the first WRF-CROCO ensemble member over (a) SA and (d) SEA. Zonal-mean precipitation anomalies for (b, e) MH$_{REF}$ and (c, f) MH$_{GS}$ over (b, c) SA and (e, f) SEA.**

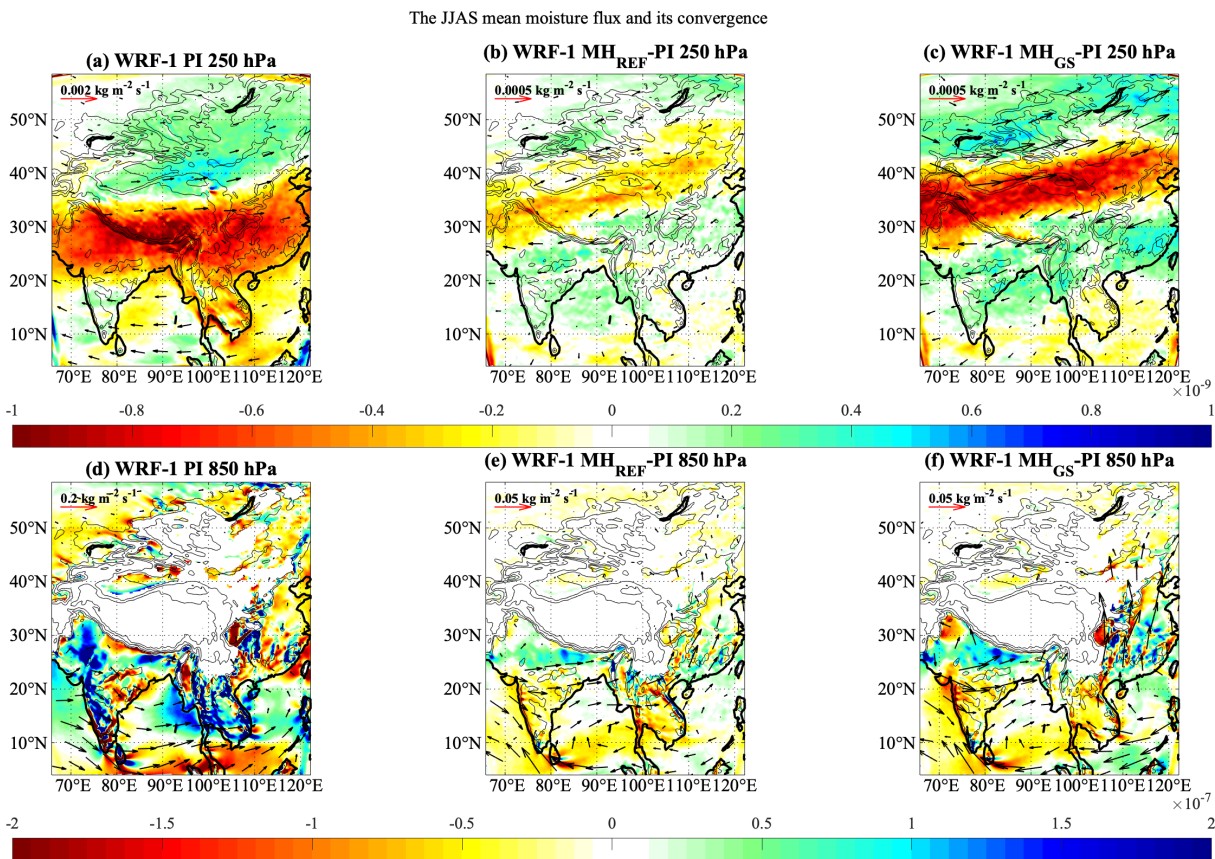

**Figure 12: Moisture flux (vector, kg m$^{-2}$ s$^{-1}$) and its convergence (shaded, kg m$^{-3}$ s$^{-1}$), convergence = blue (moisture sink), divergence = red (moisture source) for the PI simulations of the first WRF-CROCO ensemble member at (a) 250 hPa and (d) 850 hPa. Anomalies of moisture flux (vector, kg m$^{-2}$ s$^{-1}$) and its convergence (shaded, kg m$^{-3}$ s$^{-1}$) for (b, e) MH$_{REF}$ and (c, f) MH$_{GS}$ at (b, c) 250 hPa and (e, f) 850 hPa. The topography contours of 500 m, 1000 m, 2000 m and 4000 m are also shown.**