# Peer review of "Mid-Holocene monsoons in South and Southeast Asia: dynamically downscaled simulations and the influence of the Green Sahara"

_Climate of the Past, 2021_

## Author Comment (AC1)

**Response to the Comments of Referee 1 on the paper "Mid-Holocene monsoons in South and Southeast Asia: dynamically downscaled simulations and the influence of the Green Sahara" by Yiling Huo, W. R. Peltier and Deepak Chandan**

We thank the referee for his/her valuable comments on the content of our manuscript and his/her suggestions for improving the document. Following the reviewer's suggestions and comments, we have carefully revised our manuscript. We believe that the revised version satisfactorily addresses the referee's questions and concerns. In this reply, we respnd to the issues, raised by the referee point by point. Our responses to the individual comments are shown in red text following the comments in black. For convenience, the modifications made to the text will also be shown in red.

The authors use dynamically downscaled simulations to assess the impact of a vegetated Sahara on the South Asian and Southeast Asian monsoon region under mid-Holocene (MH) greenhouse gas (GHG) and orbital conditions. They couple the regional climate model WRF with the regional ocean model CROCO and drive this coupled model by output of the global model UofT-CCSM4 GCM. An ensemble of experiments is conducted using different convection schemes in WRF and different PI and MH boundary conditions.

Due to a better representation of the complex orography in South and Southeast Asia, the regional precipitation and temperature distributions are resolved in more detailed in the regional model. The MH forcing leads to an enhancement of the monsoon systems and an increase in precipitation in northern South and Southeast Asia. On the Indo-Chinese Peninsula and on the Tibetan Plateau, precipitation is rather decreased. The MH forcing furthermore leads to shifts in the monsoon season. In both areas, the monsoon onset is delayed and the withdrawal is postponed. Anomalies due to the MH forcing are generally more pronounced in the regional model than in the global model and show a better agreement to pollen-based reconstructions. However, both models are not able to capture the reconstructed mid-Holocene precipitation pattern in South China and the dry central Asian regions.

The incorporation of a vegetated Sahara enhance the precipitation response to the mid-Holocene orbital and GHG forcing and generally leads to a positive precipitation anomaly in South India, along the northern flank of the Tibetan Plateau and in North China beeing more in line with the pollen-based reconstructions. In addition,

the simulations with Green Sahara only show a sligth shift in the monsoon season compared to pre-industrial times.

The authors have taken great effort to compare the regional and global model simulations. They visualise their results with many, easy-to-read illustrations, which are described in detail and comprehensibly in the text. Unfortunately, however, the analysis rarely goes beyond these descriptions. Results are not quantified and not analysed in detail. Also, with the large number of illustrations and descriptions, the main guiding question about the influence of the green Sahara on the monsoons in the regional model is lost. I also miss a comparison with results of model studies (regional and global models) that have already been carried out on the Asian monsoon during the mid-Holocene. Since the core question is very interesting and the study could make a major contribution to better understanding and quantifying the interactions between the West African and Asian monsoons, I still recommend considering publishing the manuscript. However, major revisions are needed.

Many thanks to the reviewer for this positive feedback, which we appreciate. We agree that the readers will likely find our paper new and interesting.

The referee also suggests that the importance of the main guiding question concerning the influence of the green Sahara on the monsoons in the regional model may have been somewhat obscured by the large number of illustrations and descriptions. In the revision, we have reduced the number of figures from 12 to 11 and, for the ones we keep, we provided a more quantitative presentation of our results. We have also moved description of the sensitivities of our results to the various cumulus parameterization schemes employed in our physics mini-ensemble into the appendix. In the main text, only ensemble means (both temperature and precipitation) from the simulations are discussed.

The referee seems to be incorrect in suggesting that we have not adequately compared our simulations to previous results of model studies on the Asian monsoon during the mid-Holocene. In our discussion significant such discussion will be found (lines 196, 213-214, 223-224, 300-301, 324-325). Has the referee missed these? If there are specific references that the referee believes we have missed, it would have been more helpful to have listed them.

Main comments to the authors:

a) I understand that the monsoon precipitation distributions may strongly be influenced by the convection scheme in the model, since most rainfall stems from convective cloud cluster. However, in the context of this study, a comparison of the simulations with the different convection schemes seems to me to be too extensive. It is more a ‚ disruption ‘ than a significant contribution to underline the core message. Perhaps one could simply discuss an ensemble mean from the simulations in the main text and, for example, include the uncertainties in plots about the precipitation mean over the two regions. A comparison of the different simulations could then be presented in the supplement or appendix. Omitting the comparison would also help the paper to focus more on the main question.

We thank the reviewer for this suggestion and we have moved description of the sensitivities of our results to the various cumulus parameterization schemes employed in our physics mini-ensemble into the appendix. In the main text, only ensemble means (both temperature and precipitation) from the simulations are shown and discussed but that didn't change any of the major conclusions.

b) It is useful to compare the results with palaeo-reconstructions. Since the pollen-based reconstructions cover a larger spatial variability, it makes sense to concentrate on these reconstructions and not to discuss the cave records. On the one hand, it is still not entirely clear what the cave records recorder at all (whether changes in wind direction or changes in precipitation), and on the other hand, they are located very unfavourably, precisely on the border between positive and negative anomalies in the model. In the meantime, there is also a new pollen-based data set by Herzschuh et al. (2019) that mainly covers China. It would be interesting to see whether the deviations from the models to the reconstructions also show up in a comparison with these new reconstructions. Please use a metric to quantify your findings. Just per eye it can hardly be seen that e.g. the regional model fits better to the reconstructions than the global model for Mhref.

(reference: Herzschuh, U. et al.: Nature Communications, 10: 2376, 2019 doi:10.1038/s41467-019-09866-8).

Thank you for suggesting we integrate the new palynological data into our paper. We have compared the results with data set by Herzschuh et al. (2019) in the revision. We also used Mean Relative Error (MRE) to quantify the goodness of fit between model results and reconstructions:

"To quantify the fit to the reconstructions, Mean Relative Error (MRE) is calculated using the following Eq(1):

80  $$MRE = \frac{1}{n}\sum_{i=1}^{n}\frac{|p_i - d_i|}{|d_i|},$$  (1)

where n is the number of data points, di stands for the proxy data, pi for the model prediction. Considering all the points that have proxy data (Bartlein et al., 2011) in the WRF domain, dynamical downscaling reduced MRE of the global model by 12%. For the points north of the TP, both the GCM and the downscaled simulation fail to simulate the same sign of precipitation anomalies as indicated by the proxy data. Considering only the
85  points south of 40° N, MRE of downscaled simulation is 35 % smaller than that of the GCM. The pollen-based data set by Herzschuh et al. (2019), however, suggests much larger precipitation enhancement during the MH especially over East China and is therefore less consistent with both global and regional model results."

Regarding the usefulness of the cave data, previous modelling results (Pausata et al., 2011; Lewis et al., 2010) and analyses of modern-day instrumental records of $\delta^{18}O$ and precipitation over Asia (Dayem et al., 2010)
90  suggest that the large-amplitude precessional scale variability in the $\delta^{18}O$ in speleothems is owing to precessionally forced changes in the strength of the Indian monsoon and monsoon rainfall in south of China that are in phase and oxygen-isotope records in speleothems are often interpreted as an index for the Asian monsoon intensity. Thus, we choose not to remove the discussion on the cave records in the revision.

Dayem, K. E., Molnar, P., Battisti, D. S. & Roe, G. H. Lessons learned from oxygen isotopes in modern
95  precipitation applied to interpretation of speleothem records of paleoclimate from eastern Asia. Earth Planet. Sci. Lett. 295, 219‑230 (2010).

Lewis, S. C., LeGrande, A. N., Kelley, M. & Schmidt, G. A. Water vapour source impacts on oxygen isotope variability in tropical precipitation. Clim. Past 6, 325‑343 (2010).

Pausata, F., Battisti, D., Nisancioglu, K. et al. Chinese stalagmite $\delta^{18}O$ controlled by changes in the Indian
100  monsoon during a simulated Heinrich event. Nature Geosci 4, 474‑480 (2011). https://doi-org.myaccess.library.utoronto.ca/10.1038/ngeo1169

c) I think you could reduce the number of figures. For instance, you could show the SST and continental surface temperatures in one plot (Fig. 4 + 5). You could show the topography of both models together with the names of the geographical regions. Please think about which plot is really necessary and which do not help to underline what you want to say in your paper.

In accordance with the referees' wishes, we have now combined Figs. 4 and 5 to Fig. 4 in the revision and reduced the number of figures from 12 to 11.

JJAS continental surface air temperature and SST anomalies

[Figure]

**Figure 4: JJAS SST (contour interval 0.3° C) and continental surface air temperature anomalies (in ° C) for MHREF in (a) UofT-CCSM4 and (b) WRF. (c) shows monthly continental air temperature anomalies over SA (solid) and SEA (dashed). Shifts in calendar are not accounted for, i.e. the model calendar is used for the calculation of all anomalies. The topography contours (black) of 500 m, 1000 m, 2000 m and 4000 m are also shown in (a, b).**

d) The paper would benefit on a detailed discussion which processes are connecting the Green Sahara and South and Southeast Asia. Please already summarize in the Introduction, why the land-surface in North Africa may affect the Asian monsoon, how this teleconnection work and which dynamical circulation systems may be involved. To me it is e.g. not clear, why a greener land-surface outbalances the monsoon season shifts seen in

the MHref simulation. It would e.g. be helpful to show and discuss the precipitation pattern and the atmospheric circulation in the global model for the entire region, North Africa + South/Southeast Asia.

We added a sentence in the introduction to discuss how the land-surface in North Africa may affect the Asian monsoon:

"Pausata et al. (2017) and Sun et al. (2019) pointed out that the presence of GS conditions in northern Africa shifts the Walker Circulation westward through changes in equatorial Atlantic SSTs and warmed the Indian Ocean, which enhances the SAM and SEAM."

We also show and discusse the precipitation pattern and the atmospheric circulation in the global model for Northern Africa and SA and SEA now:

"In the global model, the GS enhances the northward expansion of the North African monsoon (Fig. 9c). Under the GS boundary conditions, the low-level westerlies over the northern Indo-Pacific Ocean becomes weaker than in $MH_{REF}$ (Fig. 9f), which decreases the upwelling and hence increases the SSTs of that region(Fig. S1), favouring more evaporation. Anomalous easterlies induced by GS over the North Pacific carry more moisture into SA and SEA, intensifying the monsoon precipitation there."

[Figure]

**Figure 11: 250 hPa winds (vector, m s⁻¹) and precipitation (shaded,mm day⁻¹) of the WRF-CROCO ensemble mean for (a) the PI simulations and anomalies in (b) MH_REF and (c) MH_GS. 850 hPa winds (vector, m s⁻¹) and SST (shaded, ° C) of the WRF-CROCO ensemble mean for (d) the PI simulations and anomalies in (e) MH_REF and (f) MH_GS. The topography contours of 500 m, 1000 m, 2000 m and 4000 m are also shown.**

e) The Introduction is very detailed, but you present a lot of information that is not really necessary to understand your paper (at least one has the feeling that it does not help to understand the paper). I recommend to re-structure the Introduction and pushing the individual parts more towards the main topic. For instance, in the first part (ll. 30 to 42) you stress the importance of the Tibetan Plateau on the Asian monsoon. Afterwards you talk about the population. I think, it would be more target-oriented to connect the importance of the Tibetan Plateau with the need to use a high spatial resolution in climate models to better represent the effect of the Tibetan Plateau on the monsoon. In global models, the Plateau is usually very flat, so why should global models capture the effect of the Plateau on the regional circulation? And this is one reason why it is so important to downscale the simulation.

Try to shorten the Introduction by beeing more precise and always keep your main topic in mind. You want to „convince " everybody that it is necessary to use regional models to analyse and understand the effect of a Green Sahara on the South and Southeast Asian monsoon. It is also important to highlight the advantages of the regional model for analysing the effect of the Green Sahara on the South and Southeast Asian monsoon.

150     We have shortened the Introduction by about 20%.

We have moved lines 35-42 in the original manuscript to the third paragraph, where we connected the importance of the Tibetan Plateau with the need of high resolution climate simulations in lines 74-86:

"In addition, the strengths of SAM and SEAM circulations and their onset, maintenance and withdrawal are to a significant extent influenced by the contiguous Tibetan Plateau (TP) that serves an elevated heat source for the
155     atmosphere in summer that intensifies the thermal contrast between the continent and ocean in the region influenced by the Asian monsoons (Wu et al., 2007). However, due to the coarse horizontal resolutions of the global models, the TP, as well as the local mountains over SA and SEA, is poorly represented in GCMs. As a result, GCMs are incapable of realistically capturing local-scale atmospheric circulation and precipitation processes, which are strongly influenced by the orography. In order to fill the need for high quality climate
160     information on regional scales while maintaining the computational tractability of the problem, this study employs the same dynamical downscaling pipeline described in Huo and Peltier (2021), to dynamically downscale MH global simulations."

f) Some sentences are really long. Please try to keep sentences short (e.g. ll 13.-17)

Thank you for this suggestion. We have shortened the long sentence in lines 13-17 to two separate sentences:
165     "In order to more accurately capture important regional features of the monsoon system in these regions, we have completed a series of regional climate simulations using a coupled modeling system to dynamically downscale MH global simulations. This regional coupled modeling system consists of the University of Toronto version of the Community Climate System Model version 4 (UofT-CCSM4), the Weather Research and Forecasting (WRF) regional climate model and the 3D Coastal and Regional Ocean Community model
170     (CROCO)."

g) It is often annoying when too many methods are not explained, but instead reference is made to other articles. Please think about explaining the main methods and giving essential informations on the models directly in this paper.

In the absence of specific details it is difficult to respond to this.

175     Section 2 and Fig. 2 give details of the primary method (dynamical downscaling) employed in this paper. Three component of the coupled model system (WRF, CROCO and UofT-CCSM4) are introduced in section 2. The major features of the MH simulations ($MH_{REF}$ and $MH_{GS}$) are also provided in this section.

However, in the revision, we have added a sentence regarding the global model UofT-CCSM4 in lines 148-149:

"It is based on the standard CCSM4 model (Gent et al., 2011) with modifications of the ocean component for
180     paleoclimate simulations."

Minor comments:

L 22: Decreased surface temperatures during mid-Holocene monsoon seasons may to a large part also result from the evaporative cooling of the surface due to enhanced precipitation.

Since our paper mainly focuses on the simulation of MH monsoon precipitation over SA and SEA and the
185     reasons for temperature changes were barely investigated, this sentence has been removed from the abstract.

LL81-92: This method part could be shifted to the end of the Introduction. It disturbs the story here.

We have now moved this sentence to section 2.

L.93: The Green Sahara is not only a , climate difference '.

In accordance with the referees' wishes, we have now changed this sentence to "During the MH, northern Africa was considerably wetter than today and was covered to a great extent by a mixture of shrubland, grassland, trees, and wetlands, —namely the existence of a Green Sahara (GS; Pausata et al., 2020; Holmes and Hoelzmann, 2017; Chandan and Peltier, 2020)."

L.155: It is not clear if you name the regional simulations or global simulations or both with MHRef.

We added a sentence at the end of section 2 to make this clear: "In the following analysis, the set of MH experiments (both global and regional) including no specific land surface changes over north Africa is referred to as the reference MH simulation and denoted as $MH_{REF}$, while the other set of MH simulations which incorporates GS boundary conditions is referred to as $MH_{GS}$."

L. 159: I somehow miss a description of the land-surface conditions in Asia. Are they also prescribed according to mid-Holocene climate conditions? Does the global model includes dynamic vegetation? East Asia is also greener during mid-Holocene and this also affects the Asian monsoon circulation.

Vegetation cover over Asia are prescribed in the model and pre-industrial land cover was used. We have added "a vegetation prescribed to PI values" in line 155 to make it clear.

The suggested experiment using interactive vegetation or prescribed MH vegetation is interesting and would provide further improvements in reconstructions of the MH Asian monsoon systems, and we will look into that in our future studies.

L. 186-189: You could check if SST records are available for the region and if they indicate the same pattern

We have added comparisons with proxy records in lines 186-188 and 192-194:

"Regional Mg/Ca (Banakar et al., 2010; Govil and Naidu, 2010) and alkenone (Böll et al., 2015) indicate 0 to 1° C cooling in the northern and eastern Arabian Sea during the MH, which is consistent with both global and regional model results."

"Proxy records also suggest slight warming off the coast of south India (Saraswat et al., 2013; Gaye et al., 2018), which is more consistent with the results of CROCO."

L.203: Please explain!

The negative temperature anomalies during spring and winter are a direct consequence of the reduced solar insolation. The cooling during early summer is likely related to the increased reflectance of shortwave flux at high levels from the greater cloud cover and increased surface evaporation due to enhanced precipitation. However, we chose not to extensively explain the reasons of the MH temperature anomalies here in the manuscript since our main focus is the SAM and SEAM precipitation.

L.211-213: , most of SA experiences wetter climate... 'In the plot most regions are yellow which means reduced precipitation during MH.'"

We are referring to the WRF results (Fig. 5b) not the GCM results (Fig. 5a) here. We have now added "(Fig. 5b)" to the text to make it clearer.

L.221: , substantial differences between global and regional model...attest to the importance of high resolution modeling···. 'Both models more or less agree to the reconstructions, but it is not clearly visible which model performs better.

We now use Mean Relative Error (MRE) to quantify the goodness of fit between model results and reconstructions in the revised manuscript (see under general comment b).

LL.224-228:Why do increased temperatures downstream of the monsoon circulation result in more precipitation, please explain!

We have removed this sentence here and the cause of precipitation increase is explained under general comment d.

L.239: Please discuss the change in East Asian monsoon circulation and its effect on the precipitation in East China.

The suggested analysis is interesting and would provide additional information concerning the East Asian Monsoon, but we feel that it falls outside the scope of this study since our study focuses upon SAM and SEAM.

L.244-245: it's , Fig.6a and 6b ʻ

We apologize for this error, and we have corrected the text as suggested.

L.245: Speleothems do not always recorder total precipitation.

This comment has already been addressed under general comment (b).

L.260-272: I would delete this part or move it to the Appendix.

This comment has already been addressed under general comment (a).

L.273: It's Figs. 6c and 7f.

We apologize for this error, and we have corrected the text as suggested.

L.308: Please explain the consequences of a reduced cooling over the northeastern Arabian Sea and southern BOB.

We have added the following sentence here to explain the consequences of SST increase here:

"The increase in SST favors more evaporation over the Arabian Sea and BOB, thus contributing to the SAM and SEAM precipitation increase."

L. 356: Please also discuss the large-scale circulation, including Northern Africa.

This comment has already been addressed under general comment (d).

L. 436: The changes in precipitation as response to the Green Sahara forcing may also feed back to the South and Southeast Asian monsoon circulation. Please comment on this.

We now added the following discussion to the revised manuscript:

"The precipitation increase as response to the GS forcing is associated with a drop in surface temperature over SA and SEAM (Fig. 7), which reduces the sensible heat flux. On the other hand, the substantially increased SAM and SEAM precipitation leads to a release of latent heat, which warms the middle and upper troposphere and adds to the temperature difference between land and ocean (Fig. 7), thus driving stronger winds and moisture advection, which in term leads to enhanced precipitation."

Fig. 1: You do not really need this figure since you hardly explain it

We disagree concerning the role of Fig. 1 and have decided to keep it. This comment seems somewhat gratuitous. We have now added reference to Fig.1 whenever our analysis mentioned a new geographical location to help the reader to appreciate the geographical setting of our analyses.

Fig. 3: The black contours are difficult to see. The monsoon circulation is also determined by the cross-equatorial temperature gradient and the SSTs in the Southern Indian Ocean. Please explain, if this fact affects your results infered by the regional model that does not include these areas.

We have increased the weight of the black contours in the revision:

[Figure]

Topography contours and outline of the WRF and the CROCO domains

Surface height (m)

-105.0    1080.0    2265.0    3450.0    4635.0    5820.0

**Figure 3: Shaded topography along with the outlines of the (shaded region) WRF and (white rectangle) CROCO domains. The two black rectangles denote the regions used to calculate spatial averages over SA and SEA. Major rivers and lakes are shown in grey contours and selected topographic heights in thin black contours.**

We agree with the referee that SSTs in the Southern Indian Ocean can also affect the Asian monsoon circulation, but extending the domain of the regional model southward to include the entire Southern Indian Ocean can make the integration computationally more expensive. Besides, the Southern Indian Ocean has less complex coastlines than the northern part and thus the added value of higher resolution is expected to be relatively smaller.

Fig. 8c) It seems that in both regions the MHGS simulations reveal higher temperatures year round (or at least during most of the year) compared to the MHREF. Please explain why and how this affects the precipitation distribution.

We added a sentence to explain the reason for the temperature increase:

280     "Such warming is directly related to the increase in the SSTs since regions with the most pronounced warming (west coast of SA and south and west SEA) also have the largest increase in SST."

As to the link to precipitation distribution, we noticed that regional precipitation and temperature anomaly distribution patterns seem to be conversely related: regions with relatively smaller precipitation increase (he west coast of SA and south and west SEA) also have larger temperature increase and vice versa. Such
285     correlation is likely due to the fact that increased precipitation is associated with increased cloud fraction and albedo, which lead to cooler temperature. Given that the magnitude of this temperature change over $MH_{REF}$ is quite small (~ 0.2°C in JJAS), it is difficult to draw definitive conclusions on the effect of this warming on precipitation here.

Fig. 9c) Why is the precipitation increased in MHGS in the post-monsoon season. It would be helpful to show
290     an anomaly plot MHGS-MHREF.

We have now added two dotted lines in this figure to show the $MH_{GS}$-$MH_{REF}$ anomalies over SA and SEA.

JJAS precipitation anomalies

[Figure]

Figure 9: JJAS average precipitation anomalies (mm day-1) for MH$_{GS}$ in (a) UofT-CCSM4 and (b) WRF ensemble mean. Monthly precipitation anomalies (mm day-1) for MH$_{GS}$ in UofT-CCSM4 and four physics ensemble members over (c) SA and (d) SEA. Shifts in calendar are not accounted for, i.e. the model calendar is used for the calculation of all anomalies. The topography contours of 500 m, 1000 m, 2000 m and 4000 m are also shown in (a, b).

Fig. 11E: heading: it is SEA instead of SA

We apologize for this error, and we have corrected the text as suggested.

Fig.12: Please also show the global model results and the circulation changes over Northern Africa.

This comment has already been addressed under general comment (d).

---

## Author Comment (AC2)

**Response to the Comments of Referee 2 on the paper "Mid-Holocene monsoons in South and Southeast Asia: dynamically downscaled simulations and the influence of the Green Sahara" by Yiling Huo, W. R. Peltier and Deepak Chandan**

We would like to thank the referee for his valuable comments on the content of our manuscript and his suggestions
for improving the document. Following the reviewer's suggestions and comments, we carefully revised our
manuscript. We believe that the revised version satisfactorily addresses the referee's questions and concerns. In
this reply, we seek to clarify the issues, raised by the referee, point by point. Please find the detailed response (red)
to the referee's comments (black).

I feel this is research is generally well-described and worthy of publication in Climate of the Past. The manuscript
is predominantly descriptive, though I have no issue with that.

I do, naturally, have some suggestions that I feel would improve the paper which should be considered before
publication. I do not feel that they will change any of the conclusions, but will help convince the reader of the
validity of those conclusions.

1. The region SA and SEA are only shown on Fig. 3 and never formerly defined. I find it strange that SA is a
square in the rotated grid of the RCM, meaning that it cross various latitudes over northern India. Given that you
are using regions and acronyms close to those used by the IPCC, at a minimum you should also show those. In
fact, I suggest that you deploy the AR6 regions from Iturbide et al. (https://doi.org/10.5194/essd-12-2959-2020) -
codes are provided to calculate them by the authors. It is also import to state whether you are only looking over
land, as in IPCC.

We have already stated in the manuscript "These two analysis regions are identical to the inner WRF domains in
Huo and Peltier (2020, 2021) wherein two levels of downscaling were employed." In these previous studies, we
have applied and validated the same dynamical downscaling pipeline over SA and SEA under modern conditions
and thus we would like to keep the definition of regions consistent with previous studies. Also, it is not quite
possible to use the AR6 regions from Iturbide et al. (2020) in this study as the Southeast Asia region in that study

25 covers Maritime Southeast Asia, which is outside of our WRF domain. Besides, the SA region in that study extend to the west to 60 ° E, which is very close to the west edge of the WRF domain. The WRF data there may suffer from relatively larger errors.

We now added at the beginning of section 3: "All spatially-averaged anomalies reported here are calculated over the land surface of the Indian subcontinent or mainland SEA south of the TP (the two black rectangles in Fig. 3)."

30 2. I suspect that if you replotted (some of) your figures as a raster rather than interpolated contours, the higher resolution of the RCM vs GCM will be much more obvious. (This could be done using imageshow rather contourf in python or CellFill in NCL).

Thank you for this suggestion. We have replotted figs. 4-9.

3. Might I suggest a different approach to the palaeo calendar issue. At present you do discuss this in the methods,
35 but it suddenly is mentioned in the figure caption. Firstly, I suspect there little benefit to calendar adjusting for an average over JJAS - we found there was no need for MJJAS in Brierley et al (2020). However, the calendar effect would alter the seasonal cycle time series that you present. If instead you plotted this seasonal cycles from daily data instead, then the issue of defining month is irrelevant. You must have daily resolution data from the GCM (to drive the RCM). I think that plotting from daily data would be more useful to identify shifts in Fig 11.

40 Thank you for this suggestion. We have now replotted fig. 11 using daily data.

[Figure]

**Figure 11: Climatological seasonal cycle of zonal-mean precipitation (mm day⁻¹) for the PI simulations of the first WRF-CROCO ensemble member over (a) SA and (d) SEA. Zonal-mean precipitation anomalies for (b, e) MH$_{REF}$ and (c, f) MH$_{GS}$ over (b, c) SA and (e, f) SEA.**

45    4. Please can you be more explicit about the 15 years selected to drive the RCM simulations. Obviously ENSO would influence monsoon rainfall, can you reassure the reader that a different sampling on ENSO events is not responsible for the patterns described?

We agree with the reviewer that ENSO would influence monsoon rainfall and that could be a source of uncertainty. We have now stated in the conclusion that "Also note that the reported precipitation changes are subject to

50    uncertainties associated with the parts of the global simulations that have been employed to force the downscaling ensemble. Since the precipitation results are strongly affected by large interannual variability associated with ENSO, further simulations with different initial conditions will be needed to characterize the internal variability and confirm the robustness of this result."

5. Some more information about improved representation of ocean upwelling the regional model would be useful

55    to place the SST changes in context.

These results for the SST from the regional ocean model are supported by the higher resolution ocean dynamics

captured in the regional ocean model but a detailed discussion must be left for the ongoing work as the first referee has already requested that the number of figures in the paper be reduced. Huo and Peltier (2021) also had some discussion on the improved representation of ocean upwelling in the regional model under modern condition (Fig. 10 in Huo and Peltier, 2021).

60

Huo, Y., Peltier, W. R.: The Southeast Asian Monsoon: Dynamically Downscaled Climate Change Projections and High Resolution Regional Ocean Modelling on the Effects of the Tibetan Plateau, Clim. Dyn., in press, https://doi.org/10.1007/s00382-020-05604-9, 2021.

6. Can you please provide some context of the GCM and RCM resolutions with respect to the rest of the PMIP4

65 ensemble.

We have added "Note here most PMIP4 models have a resolution of approximately 1° (Otto-Bliesner et al., 2017), which is close to that of our GCM UofT-CCSM4 and is considerably coarser than the resolution of our regional model."

7. Table 1 is rather uninformative. Either scrap it or, preferably, include more synthesis of the different convection

70 schemes.

This table has now been removed in the manuscript.

---

## Referee Report (RR1)

Review for revised manuscript CP-2021-17: Mid-Holocene monsoons in South and Southeast Asia: dynamically downscaled simulations and the influence of the Green Sahara by Huo et al.

The authors have addressed all comments and put a lot of effort into revising the manuscript. They carefully respond to all major issues raised by the two Referees.
They shorten the Introduction, reduced the number of figures and concentrate on the main question of the effect of a Green Sahara on the precipitation in South and South East Asia. Results are more explained than in the first version and are quantitatively evaluated against reconstructions. The sensitivity experiments with different convection schemes are shifted to the Appendix.
The manuscript is very much improved and reads much better than the first version. I agree with the publication of this revised manuscript in Climate of the Past, but still have some minor/technical suggestions:

General minor comments:
a) The introduction reads much better now, but is still very long. Please carefully look again through the paragraphs and try to further shorten it. It would also help to delete some sentences… For instance, in L 41 you state that the MH insolation was different from present-day. In the sentence afterwards you further describe this. This sentence (starting with During the MH… ) would be enough to understand the main background. The paragraph starting at L 45 with the reconstructions is very long and it does not really help to understand your paper. You could simply say, that the insolation changes intensified the NH summer monsoons (orbital monsoon hypothesis) and that palaeo-reconstructions generally confirm this view (different references…) and than go on with the "detailed knowledge is still..."
I have the same feeling with other paragraphs, that there is just too much information given that is not necessarily relevant.

b) I have to admit that I was not precise enough in my comment on the comparison with other studies. My apologies. The authors do indeed draw references to other studies. I'm just wondering if there aren't already other studies with regional models to compare the results with. For India, I remember a study with HIRHAM (Polanski et. al. 2012), dealing also with the mid-Holocene climate.

*Reference: Polanski, S., Rinke, A., Dethloff, K., Lorenz, S. J., Wang, Y., & Herzschuh, U. (2012). Simulation and comparison between mid-Holocene and preindustrial Indian summer monsoon circulation using a regional climate model. The Open Atmospheric Science Journal, 6, 42-48. doi:10.2174/1874282301206010042.*

c) Unfortunately, there are very few reconstructions for South Asia. However, one could for example compare the model results with the semi-quantitative moisture reconstructions of Wang et al. 2010. I don't know if that dataset is available, though.

d) Regarding the quantitative comparison with reconstructions: It would be helpful to include a Table, showing all MRE values for the $MH_{ref}$ and $MH_{GS}$ simulations (regional and global model).

Specific comments:

L 23: "SA" is not defined before

L 31: "monsoon" instead of "monsoons"

L 36: Do you really mean 'Additionally' or should it be 'Therefore'

L43: 'altered' instead of "enhanced", during winter NH insolation is reduced during 6ka

L44: the 20W/m², is it a mean over JJAS?

L76: A nice overview of the AHP is given in: Claussen, M., Dallmeyer, A. & Bader, J. (2017). Theory and modeling of the African humid period and the green Sahara. In *Oxford Research Encyclopedia of Climate Science* Oxford University Press. doi:10.1093/acrefore/9780190228620.013.532

L198: Do you mean Fig. 4c?

L215: Do you mean Fig. 5b?

L220-222: Since there are no reconstructions you can not state which model is correct. Maybe the reduction in precipitation seen in the global model is correct, maybe the increase in the regional model, but who knows?

L250-L256: Please include at least a warning on the cave records. I still think that they do not recorder local precipitation (see. e.g. Lui et al, 2014, or Maher, 2008)

*Zhengyu Liu, Xinyu Wen, E.C. Brady, B. Otto-Bliesner, Ge Yu, Huayu Lu, Hai Cheng, Yongjin Wang, Weipeng Zheng, Yihui Ding, R.L. Edwards, Jun Cheng, Wei Liu, Hao Yang,*
*Chinese cave records and the East Asia Summer Monsoon, Quaternary Science Reviews,*
*Volume 83, 2014, Pages 115-128, ISSN 0277-3791, https://doi.org/10.1016/j.quascirev.2013.10.021.*

*Maher BA (2008) Holocene variability of the East Asian summer*
*monsoon from Chinese cave records: a re-assessment. Holocene*
*18(6):861–866*

L260: In South China, about 30% of the rainfall occurs in the month before the monsoon sets in. This is the problem in most GCMs, they overestimate spring precipitation and also the decrease in spring precipitation due to less insolation during spring at mid-Holocene…. The decrease in spring precip exceeds the increase in summer precip and thus, the South China is drier during mid-Holocene than today (in the GCMs)

L271: During 6ka, perihelion occurs in September, so probably the overall insolation forcing was strongest during September, which may explain the strongest signal in precipitation simulated for September…

L278: It would be helpful to explain, why WRF-CROCO is more sensitive to the insolation forcing.

L345: Do you mean Fig 11e instead of 11k?

L411-412: ,including a GS'….'influence of a vegetated Sahara'   → is the same, you can delete one of it

L430 Appendix: It would be helpful if you include 1-2 sentences on the differences in the ensemble members and why you are performing ensemble simulations (It is in the method part, but I think it is helpful to repeat it here)

Fig.11: I think, the headings of the sub-figures are not correct. The Figures are mixed up. Fig. b) and c) rather look like 850hPa winds, and e+f like 250hPa wind fields. Please check!

---

## Author Response (AR2)

**Response to the Comments on the paper "Mid-Holocene monsoons in South and Southeast Asia: dynamically downscaled simulations and the influence of the Green Sahara" by Huo et al.**

We thank the referee for his/her valuable comments on our revised manuscript and his/her suggestions for improving the document. Following the reviewer's suggestions and comments, we have carefully revised our

5     manuscript again. We believe that the new version satisfactorily addresses the referee's questions and concerns. In this reply, we respnd to the issues, raised by the referee point by point. Our responses to the individual comments are shown in red text following the comments in black.

The authors have addressed all comments and put a lot of effort into revising the manuscript. They carefully respond to all major issues raised by the two Referees. They shorten the Introduction, reduced the number of

10     figures and concentrate on the main question of the effect of a Green Sahara on the precipitation in South and South East Asia. Results are more explained than in the first version and are quantitatively evaluated against reconstructions. The sensitivity experiments with different convection schemes are shifted to the Appendix.

The manuscript is very much improved and reads much better than the first version. I agree with the publication of this revised manuscript in Climate of the Past, but still have some minor/technical suggestions:

15     General minor comments:

a) The introduction reads much better now, but is still very long. Please carefully look again through the paragraphs and try to further shorten it. It would also help to delete some sentences… For instance, in L 41 you state that the MH insolation was different from present-day. In the sentence afterwards you further describe this. This sentence (starting with During the MH… ) would be enough to understand the main background. The

20     paragraph starting at L 45 with the reconstructions is very long and it does not really help to understand your paper. You could simply say, that the insolation changes intensified the NH summer monsoons (orbital monsoon hypothesis) and that palaeo-reconstructions generally confirm this view (different references…) and than go on with the "detailed knowledge is still..."

I have the same feeling with other paragraphs, that there is just too much information given that is not necessarily relevant.

We have removed the part starting at line 45 with the reconstructions and now stated that "Significant changes in the strength of the in Asian monsoon during the MH have been revealed by various paleoclimatic reconstructions, such as those based on palaeoceanographic evidence (Hutson and Prell, 1980; Prell, 1984a, b; Cullen and Prell, 1984; Prell and Van Campo, 1986), Tibetan ice cores (Thompson et al., 2000), Chinese Loess Plateau deposits (An, 2000; Porter, 2001) and stalagmites (Wang et al., 2001; Dykoski et al., 2005)."

b) I have to admit that I was not precise enough in my comment on the comparison with other studies. My apologies. The authors do indeed draw references to other studies. I'm just wondering if there aren't already other studies with regional models to compare the results with. For India, I remember a study with HIRHAM (Polanski et. al. 2012), dealing also with the mid-Holocene climate.

Reference: Polanski, S., Rinke, A., Dethloff, K., Lorenz, S. J., Wang, Y., & Herzschuh, U. (2012). Simulation and comparison between mid-Holocene and preindustrial Indian summer monsoon circulation using a regional climate model. The Open Atmospheric Science Journal, 6, 42-48. doi:10.2174/1874282301206010042.

Thank you for pointing us to this study with regional models and we have added reference to it in lines 213 and 234.

c) Unfortunately, there are very few reconstructions for South Asia. However, one could for example compare the model results with the semi-quantitative moisture reconstructions of Wang et al. 2010. I don't know if that dataset is available, though.

Does the referee mean this study: Wang, Y., et al: Asynchronous evolution of the Indian and East Asian Summer Monsoon indicated by Holocene moisture patterns in monsoonal central Asia, Earth-Science Reviews 103, 135-153, https://doi.org/10.1016/j.earscirev.2010.09.004, 2010.? If so, their data are not publicly available online and most of their data points lie in China and only four data points are in northwestern India close to the Himalayas.

However, we still added some comparison to their moisture reconstructions based on the figures in their paper in lines 219-220, 231 and 234-236.

d) Regarding the quantitative comparison with reconstructions: It would be helpful to include a Table, showing all MRE values for the MHref and MHGS simulations (regional and global model).

We thank the reviewer for this suggestion and we have added a table listing all MRE values for the $MH_{REF}$ and $MH_{GS}$ simulations.

Specific comments:

L 23: "SA" is not defined before

We have now added the definition of "SA" in line 10.

L 31: "monsoon" instead of "monsoons"

We apologize for this error, and we have corrected the text as suggested.

L 36: Do you really mean 'Additionally' or should it be 'Therefore'

We have changed to "Therefore" here.

L43: 'altered' instead of "enhanced", during winter NH insolation is reduced during 6ka

We have changed this to "altered" now.

L44: the 20W/m², is it a mean over JJAS?

Yes. The average JJAS insolation increase is approximately 20 W m$^{-2}$ in our UofT-CCSM4 model.

L76: A nice overview of the AHP is given in: Claussen, M., Dallmeyer, A. & Bader, J. (2017). Theory and modeling of the African humid period and the green Sahara. In Oxford Research Encyclopedia of Climate Science Oxford University Press. doi:10.1093/acrefore/9780190228620.013.532

Thank you for this suggestion and we have added this reference in line 72.

L198: Do you mean Fig. 4c?

L215: Do you mean Fig. 5b?

We apologize for the above two errors, and we have corrected the text as suggested.

L220-222: Since there are no reconstructions you can not state which model is correct. Maybe the reduction in precipitation seen in the global model is correct, maybe the increase in the regional model, but who knows?

We have added here ", while wetter conditions were indicated by the semi-quantitative moisture reconstructions of Wang et al. (2010)" to make our statement clearer.

L250-L256: Please include at least a warning on the cave records. I still think that they do not recorder local precipitation (see. e.g. Lui et al, 2014, or Maher, 2008)

Zhengyu Liu, Xinyu Wen, E.C. Brady, B. Otto-Bliesner, Ge Yu, Huayu Lu, Hai Cheng, Yongjin Wang, Weipeng Zheng, Yihui Ding, R.L. Edwards, Jun Cheng, Wei Liu, Hao Yang, Chinese cave records and the East Asia Summer Monsoon, Quaternary Science Reviews, Volume 83, 2014, Pages 115-128, ISSN 0277-3791, https://doi.org/10.1016/j.quascirev.2013.10.021.

Maher BA (2008) Holocene variability of the East Asian summer monsoon from Chinese cave records: a re-assessment. Holocene 18(6):861–866

We have added a sentence here: "Note here Chinese cave records have been interpreted by some studies to reflect not local MH rainfall changes but upstream monsoon rainfall or rainfall source changes (Liu et al, 2014; Maher, 2008).".

L260: In South China, about 30% of the rainfall occurs in the month before the monsoon sets in. This is the problem in most GCMs, they overestimate spring precipitation and also the decrease in spring precipitation due to less insolation during spring at mid-Holocene…. The decrease in spring precip exceeds the increase in summer precip and thus, the South China is drier during mid- Holocene than today (in the GCMs)

We agree with the referee, and have added here ", except over south China, where a decrease is simulated in annual mean while JJAS rainfall is simulated to increase (Figs. 5b and 6b)".

L271: During 6ka, perihelion occurs in September, so probably the overall insolation forcing was strongest during September, which may explain the strongest signal in precipitation simulated for September…

The referee is right to point out that perihelion occurs in September during the MH, but the local insolation forcing over SA and SEA is in fact the strongest in August (Fig. 1). The insolation change in UofT-CCSM4 is approximately 24 W m$^{-2}$ in August and 16 W m$^{-2}$ in September.

L278: It would be helpful to explain, why WRF-CROCO is more sensitive to the insolation forcing.

We added two sentences here to explain why the spatially-averaged JJAS precipitation increases produced by the WRF-CROCO ensemble mean are larger than those simulated by the UofT-CCSM4:

"Such rainfall intensification is probably related to a better representation of topography as the major wet anomaly centers in the downscaled simulation lie in the local mountain ranges over SA and SEA, including the Western Ghats, the Satpura Range in northern SA and the Garo-Khasi-Jaintia range in northwestern SEA. Moreover, the warmer SSTs over the Arabian Sea also lead to more evaporation and thus contribute to the enhanced wet anomalies, especially over SA."

L345: Do you mean Fig 11e instead of 11k?

We apologize for the this error, and we have corrected the text as suggested.

L411-412: ‚including a GS'….'influence of a vegetated Sahara' → is the same, you can delete one of it

We have now deleted ", when the influence of a vegetated Sahara is taken into account".

L430 Appendix: It would be helpful if you include 1-2 sentences on the differences in the ensemble members and
why you are performing ensemble simulations (It is in the method part, but I think it is helpful to repeat it here)

Thank you for this suggestion. We have now added a sentence here:

"Four different cumulus parameterization schemes (Tiedtke, GF, BMJ, and KF) are employed in the WRF model
to form a mini-physics ensemble, which enables us to study the sensitivity of model performance to different
cumulus parameterizations and thereby to estimate the uncertainty associated with these parameterizations on the
simulated MH climate."

Fig.11: I think, the headings of the sub-figures are not correct. The Figures are mixed up. Fig. b) and c) rather
look like 850hPa winds, and e+f like 250hPa wind fields. Please check!

The headings of the sub-figures are correct. The 850 hPa wind maps use a smaller reference value (the red arrow
in the right bottom corner of each sub-figure), so the 850 hPa wind arrows in Figs. 11e and 11f appear to be longer
than the 250 hPa wind arrows in Figs. 11b and c.